



# The role of the stratospheric state in upward wave flux prior to Sudden Stratospheric Warmings: a SNAPSI analysis

Blanca Ayarzagüena[1], Amy H. Butler[2], Peter Hitchcock[3], Chaim I. Garfinkel[4], Zac D. Lawrence[5,6], Wuhan Ning[4], Philip Rupp[7], Zheng Wu[8], Hilla Afargan-Gerstman[9], Natalia Calvo[1], Alvaro de la Cámara[1], Martin Jucker[10], Gerbrand Koren[11], Daniel De Maeseneire[1], Gloria L. Manney[12,13], Marisol Osman[14,15,16], Masakazu Taguchi[17], Cory Barton[18], Dong-Chang Hong[19], Yu-Kyung Hyun[20], Hera Kim[21], Jeff Knight[22], Piero Malguzzi[23], Daniele Mastrangelo[23], Jiyoung Oh[20], Inna Polichtchouk[24], Jadwiga H. Richter[25], Isla R. Simpson[25], Seok-Woo Son[19,26], Damien Specq[27], and Tim Stockdale[24]

[1] Department of Earth Physics and Astrophysics, Facultad de CC. Físicas, Universidad Complutense Madrid, Spain

[2] National Oceanic and Atmospheric Administration (NOAA) Chemical Sciences Laboratory, Boulder, CO, USA.

[3] Department of Earth and Atmospheric Sciences, Cornell University, USA.

[4] Fredy and Nadine Herrmann Institute of Earth Sciences, The Hebrew University of Jerusalem, Jerusalem, Israel.

[5] Cooperative Institute for Research in Environmental Sciences (CIRES), University of Colorado Boulder, Boulder, CO, USA

[6] National Oceanic and Atmospheric Administration (NOAA) Physical Sciences Laboratory, Boulder, CO, USA

[7] Faculty of Physics, Ludwig-Maximilians-University Munich, Germany

[8] Department of Atmospheric and Environmental Sciences, SUNY Albany, USA.

[9] Institute for Atmospheric and Climate Science, ETH Zürich, Switzerland.

[10] Climate Change Research Centre, University of New South Wales, Sydney, Australia

[11] Copernicus Institute of Sustainable Development, Utrecht University, The Netherlands

[12] NorthWest Research Associates, USA.

[13] New Mexico Institute of Mining and Technology, USA

[14] Universidad de Buenos Aires, Facultad de Ciencias Exactas y Naturales, Departamento de Ciencias de la Atmósfera y los Océanos.

[15] CONICET – Universidad de Buenos Aires. Centro de Investigaciones del Mar y la Atmósfera (CIMA).

[16] CNRS – IRD – CONICET – UBA. Instituto Franco-Argentino para el Estudio del Clima y sus Impactos (IRL 3351 IFAECI). Buenos Aires, Argentina.

[17] Department of Earth Science, Aichi University of Education, Kariya, Japan.

[18] Space Science Division, US Naval Research Laboratory, Washington, DC, USA

[19] School of Earth and Environmental Sciences, Seoul National University, Republic of Korea

[20] National Institute of Meteorological Science, Korea Meteorological Administration, Jeju, South Korea.

[21] Department of Atmospheric Sciences, University of Washington, Seattle, WA, USA

[22] Met Office Hadley Centre, Exeter, UK.



[23] CNR-ISAC, Bologna, Italy.

[24] European Centre for Medium-range Weather Forecasts, Reading, UK.

[25] Climate and Global Dynamics Laboratory, National Center for Atmospheric Research, Boulder, CO, USA.

[26] Interdisciplinary Program of Artificial Intelligence, Seoul National University, Republic of Korea.

[27] Centre National de Recherches Météorologiques, Université de Toulouse, Météo-France, CNRS, Toulouse, France.

*Correspondence to*: Blanca Ayarzagüena (bayarzag@ucm.es)

**Abstract.** Several studies highlight the relevance of considering polar winter stratospheric information such as the occurrence of Sudden Stratospheric Warmings (SSWs) for skillful Subseasonal to Seasonal (S2S) surface climate predictions. However, current S2S forecast systems can only predict these events about two weeks in advance. A potential way of increasing their predictability is to improve the models' representation of the triggering mechanisms of SSWs. Traditional theories indicate SSWs follow sustained wave dissipation in the stratosphere, but the relative role of tropospheric versus stratospheric conditions

in the enhancement of stratospheric wave activity remains unclear.

This study aims to quantify the role of the stratospheric state in wave activity preceding SSWs by analyzing three recent SSWs: the boreal SSWs of 2018 and 2019 and the austral minor SSW of 2019, using specific sets of S2S experiments. These ensembles follow the SNAPSI (Stratospheric Nudging And Predictable Surface Impacts) guidelines and include free-evolving atmospheric runs and nudged simulations, where the zonally-symmetric stratospheric state is nudged to either observations of

a certain SSW or a climatological state. Our results show that the models struggle to capture the strong enhancement of wave activity preceding the 2018 SSW, limiting predictability beyond 10 days. In contrast, both SSWs of 2019 are better simulated, consistent with a more accurate simulation of the wave activity. The zonal mean stratospheric state does not drastically influence the upward wave activity flux or tropospheric circulation anomalies prior to these SSWs, but it has some impact on the stratospheric wave activity, although this modulation depends on the event characteristics. The boreal 2019 SSW appears

to be primarily driven by tropospheric processes. In contrast, stratospheric contributions may have also played an important role in triggering the boreal 2018 SSW and the austral 2019 SSW. Understanding these variations is key to improving SSW predictability in S2S models.

## 1 Introduction

Sudden stratospheric warmings (SSWs) are the most dramatic example of wintertime polar stratospheric dynamical variability.

They are characterized by a rapid increase of polar temperatures and a complete reversal of the climatological westerly winds in the middle stratosphere (Baldwin et al 2021). The associated effects of these phenomena span beyond the stratosphere. For instance, in the troposphere the SSW signal projects onto a negative phase of the Annular Mode that can persist for several weeks up to two months (Baldwin and Dunkerton, 2001).



Given the persistent influence of SSWs on tropospheric weather, incorporating stratospheric information into subseasonal to seasonal (S2S) forecast systems has proven beneficial for increasing the skill of S2S predictions of surface climate (e. g., Sigmond et al. 2013; Domeisen et al. 2020b). However, current S2S forecast systems can predict SSWs only about two weeks before they occur (Domeisen et al. 2020a; Chwat et al 2022). Thus, improving the predictability of SSWs within these models is crucial for obtaining better wintertime S2S surface weather forecasts. One potential strategy to achieve this is by improving the models' representation of SSW triggering mechanisms, which, in turn, requires a deeper understanding.

Traditional theories indicate that SSWs are preceded by sustained wave dissipation in the stratosphere, primarily driven by wave amplification and nonlinear wave breaking in that layer (Baldwin et al. 2021 and references herein). The main waves involved in this process correspond to ultra-long planetary (e.g. of wavenumbers 1 and 2) waves that are primarily generated in the troposphere by topography and thermal land-sea contrasts (Garfinkel et al. 2010). Thus, the tropospheric state and its effects on planetary-scale waves play a key role in the wave amplification. However, the stratosphere also exerts some control over the upward-propagating wave activity. For instance, Rossby waves can only propagate in westerly winds, meaning that tropospheric waves only reach the stratosphere from autumn to spring (Charney and Drazin, 1961). The interactions between waves and the stratospheric mean flow also influence wave amplification. As waves dissipate, they decelerate the westerly mean flow, allowing a stronger upward propagation of wave activity if there is no wind reversal (Holton and Mass, 1976). Thus, the exact cause of the wave amplification leading to the occurrence of an SSW, and specifically the relative roles of the stratospheric and tropospheric states, are still under debate (Butchart, 2022).

Some authors emphasize the importance of enhanced and persistent tropospheric wave forcing as a key factor in initiating an SSW (e.g.: Matsuno 1971). In this scenario, the development of an SSW then takes some time, since the wave activity builds up over time starting from the tropospheric source and propagating upward into the stratosphere (Cohen and Jones, 2011; Sjoberg & Birner, 2012; Schwartz and Garfinkel, 2017). In contrast, other studies have provided evidence of wave amplification occurring within the stratosphere without a corresponding enhancement of wave activity in the troposphere (e.g., Jucker 2016; Birner and Albers 2017; de la Cámara et al. 2019). This may result from two distinct mechanisms: the first is a "valve" effect of the lower stratospheric basic flow, modulating and/or channeling the wave activity flux that enters into the stratosphere (e.g., Chen and Robinson 1992; Scott and Polvani 2004, 2006; Hitchcock and Haynes 2016); the second is through resonant wave growth excited by the stratospheric flow configuration (e.g., Clark 1974, Plumb 1981, Smith 1989, Matthewman and Esler 2011; Esler and Matthewman 2011; Albers and Birner 2014). Regardless of the mechanism at work, the evolution of the stratosphere towards an SSW-favorable state is generally known as "preconditioning" (McIntyre 1982; Lawrence and Manney, 2020) and may involve changes in the stratospheric basic state not directly linked to the polar vortex.

Observational evidence supports both the tropospheric and stratospheric roles in triggering SSWs and suggests that the mechanism at play may depend on the type of SSW. Whereas anomalous tropospheric wave activity tends to precede wavenumber-1 (WN1) events (Birner and Albers 2017), wave resonant processes in the stratosphere are more likely to be involved in triggering wavenumber-2 (WN2) SSWs (Albers and Birner 2014). In any case, the main difficulty of analyzing the different triggering mechanisms in observations or comprehensive models lies in the inherent nonlinear wave-mean flow





interactions involved, which hinders efforts to distinguish cause from effect (e.g., Sjoberg and Birner 2014, de la Cámara et al. 2017). Clarifying these dynamics is crucial for improving SSW predictability.

Apart from the use of idealized models (e.g., Gerber and Polvani, 2009; Hitchcock and Haynes, 2016), nudging the atmospheric state in a certain region or layer in complex models may help to disentangle the role of the troposphere and stratosphere in triggering SSWs. Indeed, this technique has been successfully applied in climate models (de la Cámara et al. 2017) or even in S2S models (e.g. Kautz et al. 2020), although in the latter case for a different purpose. Most of these studies have used only one model, but S2S models have biases in the stratospheric state and wave activity (Lawrence et al., 2022; Garfinkel et al.

2025). Thus, the results derived from a single model study can be influenced by these biases and a multimodel approach would be more appropriate.

In this study, we aim to investigate the stratospheric role in the amplification and propagation of upward wave activity during SSWs. The use of a set of S2S experiments of the Stratospheric Nudging And Predictable Surface Impacts (SNAPSI) project (Hitchcock et al. 2022) provides a unique opportunity to achieve this goal. The SNAPSI experiments are performed with

several S2S models following the same requirements. They are designed to isolate the effects of the zonal mean stratospheric state (through nudging) on the rest of the atmosphere and in particular, on the troposphere during three SSWs: the boreal SSWs of 12th February 2018 and 2nd January 2019 and the austral minor SSW of September 2019. These events were very different in many regards. In terms of surface impacts, the SSW of 2018 (SSW2018) and the austral SSW (SSW2019 SH) led to numerous extreme surface events (Ayarzagüena et al. 2018; Kautz et al. 2020; Lim et al. 2021), while the influence on surface

weather of the boreal SSW of 2019 (SSW2019) was weak (Butler et al. 2020). The two SSWs of 2019 were predictable by S2S systems at longer lead times than the 2018 event (Butler et al. 2020; Rao et al 2020). The dynamics that preceded the three events also differed remarkably: The SSW2018 was preceded by a rapid amplification of WN2 wave activity, whereas the other two events were mainly associated with strong WN1 wave activity (Karpechko et al. 2018; Butler et al. 2020; Lim et al. 2021).

Previous studies have shown that WN2 SSWs are more difficult to forecast than WN1 events (Taguchi 2018; Domeisen et al. 2020a; Chwat et al 2022), but the underlying physical reasons remain unclear. While Taguchi (2018) performed a systematic multimodel analysis of tropospheric precursors and wave activity preceding SSWs, SNAPSI will allow us to make a step forward on this topic by investigating and isolating the influence of the stratospheric state on these triggering mechanisms depending on the type of event. In addition, SNAPSI will also contrast the ability of S2S forecast systems to reproduce each

of the three events. In particular, we first assess the ability of different S2S forecast systems to reproduce the stratospheric wave amplification and tropospheric wave structures preceding the SSWs in Section 3. Secondly, we investigate the influence of the stratospheric state on the triggering mechanisms of the three SSWs in Section 4. Given the models' low skill in predicting the SSW2018 and the associated wave activity, we perform a detailed analysis of the upward wave propagation prior to this event in Section 5.




## 2 Data and methodology

### 2.1 SNAPSI experiments

As previously mentioned, we use the set of ensemble SNAPSI experiments of seven S2S forecast systems (Table 1). These ensembles include free-evolving atmospheric runs (FREE experiment) and nudged simulations where the zonally-symmetric stratospheric state is nudged globally to either observations of a specific SSW (NUDGED) or a time-evolving climatology (CONTROL) (Hitchcock et al. 2022). The stratospheric states in these nudged ensembles are derived from 6-hourly data taken from ERA5 reanalysis at model levels (Hersbach et al. 2020). For the CONTROL ensemble, the daily climatological state is calculated as the average of ERA5 data from 1st July 1979 to 30th June 2019. The zonally symmetric states used for NUDGED and CONTROL correspond to instantaneous zonal-mean temperature and zonal wind from ERA5 at 6h intervals (Hitchcock et al. 2022). The nudging consists of a relaxation tendency towards the zonally symmetric reference state to which the flow is constrained as follows: $-\tau^{-1}(X - X_t)$, where $X$ is the field to be nudged and $X_t$ corresponds to the reference stratospheric state to which the field is constrained. The nudging is applied gradually in the vertical starting at 90hPa and reaching the full strength at 50hPa. This is expressed by modifying the timescale of the relaxation with height as: $\tau = 6h \cdot \left(\frac{p_b - p}{p_b - p_t}\right)^3$, where $p_b$= 90hPa and $p_t$= 50hPa. The choice of these vertical levels is made to avoid a direct impact on the troposphere following the recommendations of previous research (Hitchcock and Haynes 2016). The stratospheric nudging is imposed at all latitudes and longitudes equally, so that the zonally asymmetric part of the flow and thus, the wave field are not affected. This latter point is particularly important for the purpose of the present study, focused on the wave activity flux. In all cases, as mentioned in Hitchcock et al. (2022), wave activity fields in the nudged experiments are comparable to the corresponding ones in the FREE experiments. The nudging can indeed interfere with the interaction between the waves and the mean flow which can then impact the wave structures themselves, but Hitchcock et al. (2022) found this to only have significant impacts on wave amplitudes in the upper stratosphere as a result of waves propagating to higher levels in the absence of their ability to decelerate the mean flow lower down.

Each forecast integration spans 45 days, with ensemble sizes of 50 members, except for the NAVGEM model that includes 80 members. Furthermore, each experiment is initialized on two distinct dates for each SSW event: one prior to the onset of the event and another approximately three weeks before the associated surface extreme events. Since our focus here is on the mechanisms that trigger SSWs, we use the first initialization dates with a similar forecast lead time with respect to the SSW: 2018-01-25 (18 days before the event), 2018-12-13 (20 days before the event), and 2019-08-29 (17 days prior to the SSW). For further details, please refer to Hitchcock et al. (2022).

Along the study, results derived from SNAPSI experiments are compared with those obtained with ERA5.



**Table 1. List of models included in the analysis indicating their resolution and key reference.**


| Model center | S2S forecast system | Atmospheric resolution | Reference |
|---|---|---|---|
| NCAR | CESM2-CAM6 | 1.25º × 0.9º, top 2hPa | Richter et al (2022) Danabasoglu et al (2020) |
| Météo-France | CNRM-CM6-1 | TL359, top 0.01hPa | Voldoire et al (2019) |
| CNR-ISAC | GLOBO | 0.7° × 0.5º, top 0.15hPa | Malguzzi et al (2011) |
| SNU | GRIMs | T126, top 0.3 hPa | Koo et al. (2023) |
| UKMO | UKMO-GloSea6 | N216, top 85km | Williams et al. (2017) |
| KMA | KMA-GloSea6 | N216, top 85km | Walters et al. (2017) |
| ECMWF | IFS-48r1 | TCo319, top 0.01hPa | ECMWF documentation |
| NRL | NAVGEM | T359, 0.04hPa | Hogan et al. (2014) |

## 2.2 Methods

We describe here the metrics and techniques used to achieve our goals. Since our focus is on the mechanisms that trigger SSWs, these diagnostics emphasize event identification and quantification of wave activity fluxes.


*SSW identification*

SSWs in the Northern Hemisphere (NH) are identified by the reversal of the westerly zonal-mean zonal wind at 60ºN and 10hPa to easterly (Charlton and Polvani, 2007). Once an event is detected in a given realization, the search for additional SSWs within the same realization is stopped. For the Southern Hemisphere (SH) case, the observed SSW2019 SH was classified as

a minor warming with a strong deceleration of the wind but without a full reversal. Thus, the SH events are identified when the zonal-mean zonal wind at 60ºS and 10hPa weakens below 20 m s⁻¹ (as in Rao et al (2020a)).

*Diagnostics of wave activity*





Several diagnostics are used to analyze wave activity, particularly upward wave activity flux. These diagnostics include the
Eliassen-Palm flux, the zonal-mean meridional eddy heat flux averaged over 45º-75º (HF), and the refractive index.

The Eliassen-Palm flux (EP flux, $F$) (Eq.1) represents the flux of wave activity, providing insights into the dynamics of wave-mean flow interactions (Edmon et al. 1981).

$$F = (F_\phi, F_z) \tag{1}$$

where the two $F$ components are defined as follows (Andrews et al. 1987):

$$F_\phi = \rho\, a\, cos\, \phi \left[ -\overline{u'v'} + \overline{u_z}\frac{\overline{v'\theta'}}{\overline{\theta_z}} \right] \tag{1.1}$$

$$F_z = \rho\, a\, cos\, \phi \left[ \left( f - \frac{1}{a\, cos\phi}\frac{\partial(\overline{u}\, cos\phi)}{\partial\phi} \right)\frac{\overline{v'\theta'}}{\overline{\theta_z}} - \overline{u'w'} \right] \tag{1.2}$$

In these equations, $\rho$ is the density, a is the Earth's radius, $\phi$ is the latitude, $f$ refers to the Coriolis parameter, $u$, $v$, and $w$ are
the zonal, meridional and vertical wind, and $\theta$ is the potential temperature. The overbar indicates the zonal mean, the prime (') represents deviations from this mean, and the subscript $z$ denotes partial differentiation with respect to $z$. In some analyses, $u'$, $v'$, and $\theta'$ are calculated for specific wavenumbers (WN = 1, 2 and 3) using Fast Fourier Transform filters.

The EP flux information is used in three different ways in Section 5:
1. The representation of the two $F$ components as a vector illustrates the direction of meridional and vertical wave propagation on a specific day.
2. The time evolution of $F_z$ averaged over 50º-70º N at both tropospheric and stratospheric levels is displayed to study vertical wave propagation surrounding the SSW2018.
3. The $F$ budget over the stratospheric region between 100 and 10hPa and 55ºN and the pole is shown as a measure of
the resolved wave driving (Sigmond and Scinocca, 2010; Wu et al. 2019). The separate analysis of the different terms of this budget shown in Eq. (2) enables us to determine the main contribution to the weakening of the polar vortex in each experiment and their modulation by the stratospheric state.

$$Net\ F\ budget = \int_{55N}^{90N} cos\phi F_z d\phi|_{100hPa} - \int_{55N}^{90N} cos\phi F_z d\phi|_{10hPa} - \frac{1}{a}cos\phi \int_{10hPa}^{100hPa} F_\phi dz|_{55N} =$$

$$F_{100} - F_{10} - F_{55N} \tag{2}$$


$Net\ F\ budget$ is positive when there is a convergence of $F$ towards the polar region. Consistently, $F_{100}$ and $F_{10}$ are also defined as positive when pointing upward and $F_{55N}$ when directed equatorward.



As a simple metric of $F_z$, we use the zonal-mean meridional eddy heat flux (HF), defined in Eq. (3). HF is used in sections 3
and 4 as a measure of the upward extratropical wave activity flux at specific levels, with 100 hPa representing the stratosphere
and 300hPa the upper troposphere.

$$HF \ = \ \overline{v'T'} \tag{3}$$

The refractive index ($n^2$) provides insight into the direction of the wave propagation based on atmospheric conditions. It is
calculated for stationary waves using Eq. (4) (Weinberger et al. 2021):

$$n^2(\phi, z) \ = \ \frac{\overline{q_\phi}}{a\,\overline{u}} - \frac{s^2}{a^2 cos^2\phi} - \frac{f^2}{4\,N^2(\phi,z)\,H^2} \tag{4}$$

where $s, H$, and $N$ denote the zonal wavenumber, the height scale, and the buoyancy frequency, respectively and $q$ corresponds
to the potential vorticity.

All these diagnostics are computed on the available model grid data.

*Isolation of the atmospheric state's effects on SSW occurrence*

To investigate the stratosphere's role in the upward wave propagation and amplification, various approaches are employed.
The primary method consists of comparing results from the NUDGED and CONTROL experiments, where the stratospheric
state follows the atmospheric evolution of the corresponding SSW in the former and the climatological state in the latter.

Alternatively, to identify the tropospheric precursors of the SSWs, the 15 "weakest u" and 15 "strongest u" ensemble members
from the FREE experiment of each model are selected and analyzed. This selection is based on the zonal-mean zonal wind at
60º and 10hPa averaged during five days around the onset date of each SSWs (Table 2). The "weakest u" members are those
with the 15 lowest wind values, indicating either very weak westerlies or even easterly winds and thus, an SSW or weak vortex
conditions during the same dates as observed. The "strongest u" members, having the 15 highest wind values, point to a very
strong vortex. As such, the comparison of the tropospheric circulation in these two groups of ensemble members allows us to
identify the tropospheric circulation structures more likely related to the occurrence of the SSWs by affecting the wave activity
entering the stratosphere and impacting the zonal mean.




**Table 2. Time periods considered for the selection of the 15 "weakest u" and 15 "strongest u" ensemble members.**

| SSW event | Period |
|---|---|
| SSW2018 | 2018-02-12 to 2018-02-16 |
| SSW2019 | 2019-01-02 to 2019-01-06 |
| SSW2019 SH | 2019-09-18 to 2019-09-22 |

## 3 Ability of S2S systems to predict the SSWs and their precursors

In this Section, we evaluate the ability of models to predict the three SSWs, the associated wave activity and their tropospheric

precursors.

### 3.1 SSWs occurrence

As a first step, we aim to assess the models' skill in predicting the occurrence of each SSW. Figure 1 illustrates the time evolution of the zonal-mean zonal wind at 60º latitude and 10hPa (u60_10) corresponding to each SSW for ERA5 (black line) and the ensemble mean of the FREE experiment of each model (colored lines). The daily climatology of ERA5 is also

represented by the dotted black line. According to ERA5, the wind reversals took place on 12th February 2018 and 2nd January 2019 for the NH events. The central date of the SSW2019 SH is taken as 15 September 2019, when the u60_10 dropped below 20 m s$^{-1}$, as described in Section 2.2. The state of the polar night jet (PNJ) at the beginning of the simulation is different in each case. The PNJ prior to the SSW2018 was stronger than the climatology and around the climatological state for the SSW2019. In contrast, the PNJ was already weaker than the climatology in the initial state of the SSW2019 SH.

Considering that the forecast lead time with respect to the SSW occurrence is very similar in the three experiments, the models experienced the greatest challenge in predicting the SSW2018 (Fig. 1a vs Fig. 1b&c), consistent with previous literature (Butler et al. 2020; Rao et al., 2018, 2019, 2020b). In this case, the ensemble mean of nearly all models struggles even to simulate a deceleration of the PNJ, with less than 30% of the ensemble members simulating an SSW in all models except for CNRM-CM6-1 (Table 3). NAVGEM also shows a relatively high number of ensemble members with an SSW, but most of them occur

much later than in observations, i.e., late February and March, consistent with the weak values of the ensemble mean of u60_10 by that time. In contrast, the models' prediction skill for the two SSW events in 2019 is higher. For the NH event, all models except GLOBO simulate an SSW in many of the ensemble members (Table 3), but they tend to predict the event too early, as seen in the ensemble mean of u60_10 for CNRM-CM6-1 and both GloSea6 models (Fig. 1b). For the SH SSW, the deceleration of the PNJ is captured by the models over the first week. For the following days, whereas the PNJ in ERA5 continues




decelerating even faster than in the previous days, models simulate a much weaker deceleration except for IFS and CNRM-CM6-1 (Fig. 1c).

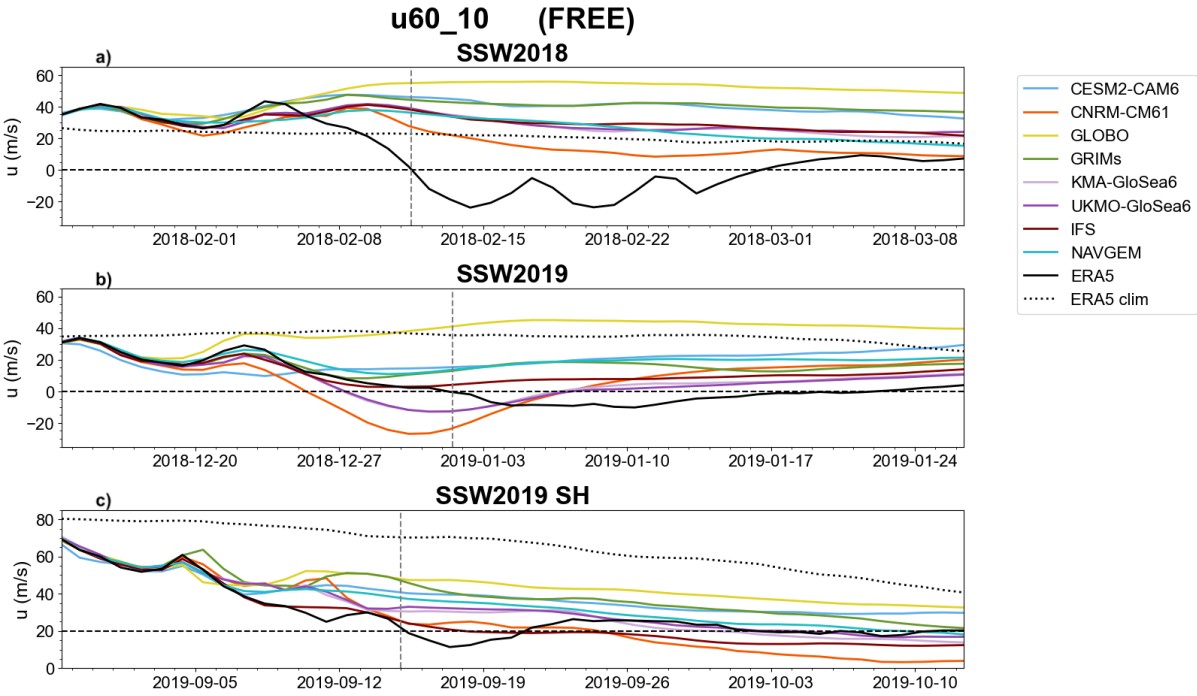

**Figure 1. Time evolution of the ensemble mean of the zonal mean zonal wind (m s⁻¹) at 60º latitude and 10hPa for each model (colored lines), ERA5 (black line), and ERA5 climatology (black dotted line) during the a) SSW 2018, b) SSW 2019 and c) SSW 2019 SH. The dashed vertical line represents the timing of the fulfillment of the SSW criterion.**

**Table 3. Percentage of ensemble members (%) that predict an SSW during the 45 days of the FREE experiment (first initialization).**

|  | CESM2-CAM6 | CNRM-CM6-1 | GLOBO | UKMO-GloSea6 | KMA-GloSea6 | GRIMs | IFS | NAVGEM |
|---|---|---|---|---|---|---|---|---|
| **2018-01-25** | 10% | 62% | 0% | 26% | 26% | 4% | 18% | 44% |
| **2018-12-13** | 48% | 100% | 2% | 96% | 94% | 52% | 80% | 46% |
| **2019-08-29** | 28% | 90% | 4% | 64% | 68% | 26% | 90% | 47.5% |

**3.2 Upward wave activity flux**

Most of the model issues in simulating the three SSWs are consistent with model difficulties in reproducing the upward-propagating wave activity, as shown in the heat flux (HF) prior to the events in both the troposphere (300hPa) and stratosphere





(100hPa) (Figs. 2-4). In ERA5, SSW2018 was preceded by a short but intense WN2 burst of wave activity in both the stratosphere and troposphere that started on 5th February (black line Fig. 2a&b). However, the ensemble mean of HF in the
forecast systems misses this strong enhancement of WN2 wave activity at both levels, particularly from 7[th] February (Fig.2). This explains the general lack of deceleration of the PNJ from that date seen in Fig. 1a. The best performing forecast system is the CNRM-CM6-1: this is the only model with more than half of ensemble members simulating an SSW (62%) and, consistently, it shows the strongest wave activity in the first days of the big pulse, particularly at 100hPa (Fig.2b). Furthermore, it is the only model that reproduces a preceding secondary peak of wave activity on 2[nd]-5[th] February at 300hPa.

The SSW events of 2019 were both initiated by a moderate but persistent amplification of WN1 wave activity that is, in general, captured by the models (Fig. 3a&c and 4a&b). The results agree with the higher forecast skill for these two events. However, there are still model deviations from ERA5. The WN3 wave activity seems to play a relevant role for the occurrence of the NH event (Butler et al. 2020, Rao et al. 2019), but models struggle to simulate the WN3 burst around 4-5 days before the SSW onset date (Fig. 3b&d). In particular, the forecast systems either miss it or simulate a weaker anomaly and earlier than in ERA5.
The latter happens in CNRM-CM6-1 and both GloSea6 models, which are the models with the highest SSW forecast skill, but with a too early SSW date (Fig. 1b). This may suggest that the simulation of the WN3 pulse might be linked to the forecasted timing of the SSW in each model. In this sense, Rao et al. (2019) and Butler et al. (2020) showed that during this event, the polar vortex first decelerated and shifted out of the pole and then split. The wind reversal happened when the vortex split took place and has been linked to the peak of WN3 wave activity following the WN1 persistent burst (Rao et al. 2019). As for the
SH event, in general, the WN1 burst is weaker in models than in the reanalysis, explaining the weaker modeled deceleration of the PNJ (Fig. 4a&b). Further, the models capture the WN1 magnitude in the stratosphere reasonably well initially, but they fail to simulate its long persistence (Fig. 4a). IFS is the model that best performs at both levels, particularly in terms of the persistence of the WN1 burst in both layers. This is also the model with the closest representation of the PNJ deceleration to observations (Fig. 1c).





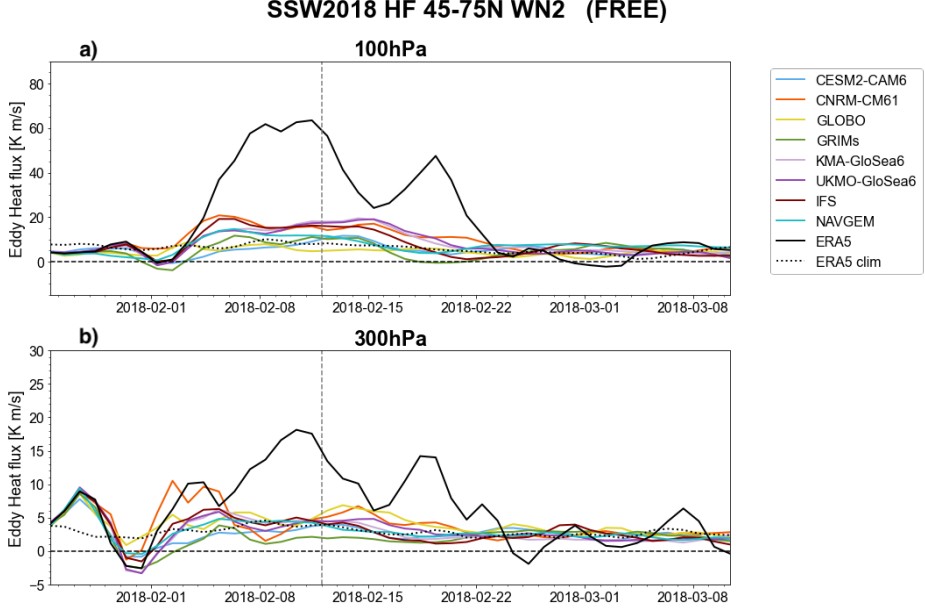

**Figure 2. Time evolution of the ensemble-mean of the extratropical eddy heat flux (K m s$^{-1}$) at (a) 100hPa and (b) 300hPa during the SSW 2018 for the WN2 component. The dashed vertical line denotes the date of the SSW.**

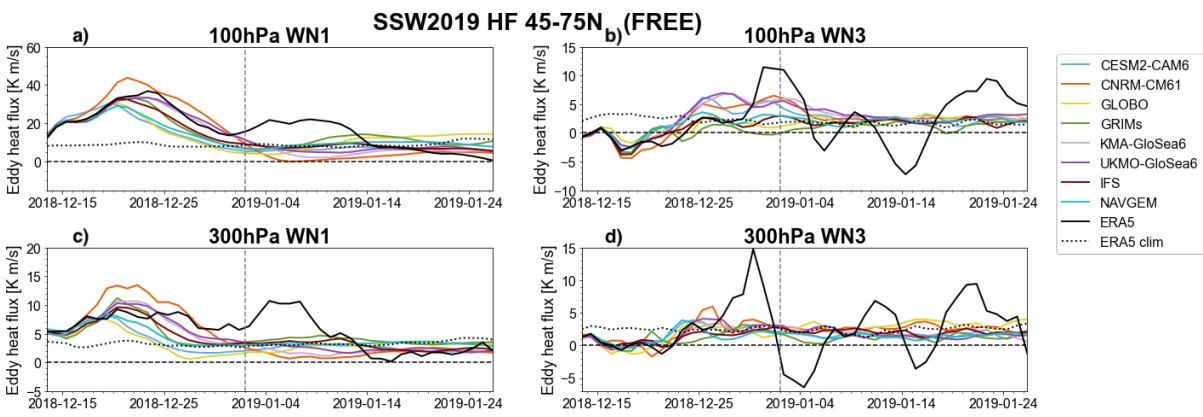

**Figure 3. Time evolution for the NH SSW2019 of the ensemble mean of the extratropical eddy heat flux (K m s$^{-1}$) at 100 hPa for (a) WN1 and (b) WN3 wave components. (c) and (d) same as (a) and (b) but for the heat flux at 300hPa. The dashed vertical line denotes the date of the SSW.**




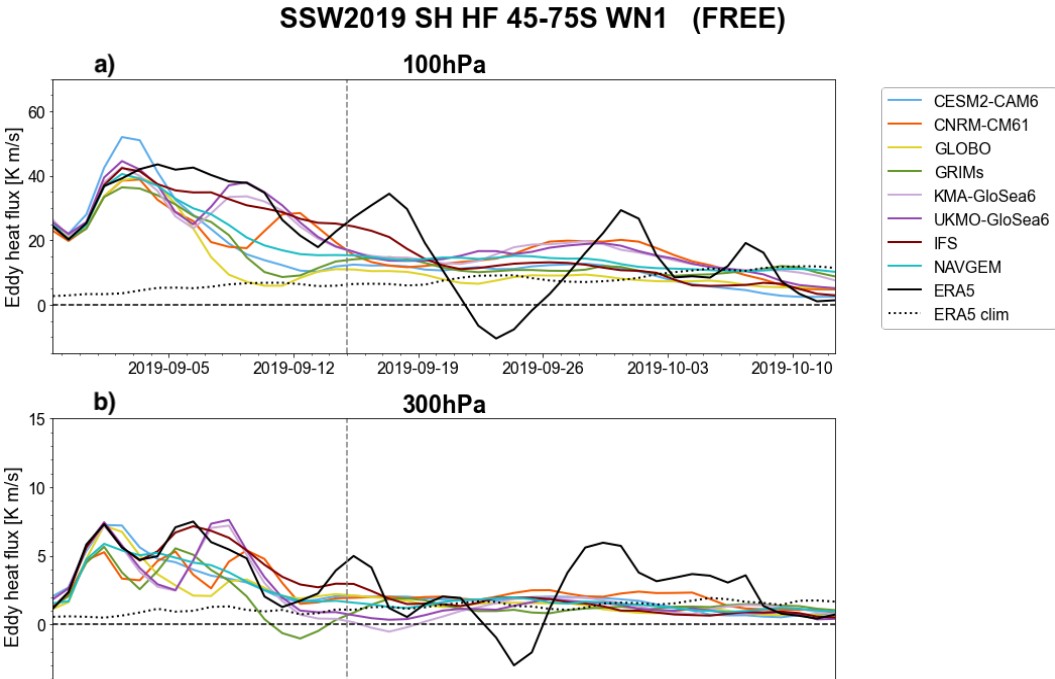

**Figure 4. Same as Figure 2 but for the SSW 2019 SH and the WN1 wave component. The heat flux values are multiplied by -1 to be**
**comparable to the NH cases.**

### 3.3 Tropospheric circulation preceding SSWs

In previous subsections, we have identified a link between wave forcing and the deceleration of the PNJ, particularly, in relation to model biases. Given that the main sources of the wave activity are generally located in the troposphere, we extend our analysis to examine tropospheric conditions during the development of the three SSWs in the FREE experiment. Specifically,
we focus on the period with intense tropospheric wave-driving for each of the three SSWs of study. We investigate potential differences in tropospheric circulation between members with a weak versus strong PNJ at the time the SSW was observed in ERA5. This allows us to determine whether the tropospheric state was then different between these two members groups and so played a role in the occurrence of SSWs. Further, this tropospheric analysis helps us identify tropospheric structures potentially influencing the likelihood for an SSW. To do so, Figs. 5-7 show the composite maps of differences in Z500
anomalies between the 15 "weakest u" FREE members and the 15 "strongest u" FREE members (Table 2). These composite maps are calculated for the days corresponding to the strongest burst of upward-propagating wave activity in the troposphere (300hPa) in ERA5 (3-13 Feb 2018; 18-30 Dec 2018; 1-10 Sep 2019).





# FREE SSW 2018 (3-13 Feb 2018)



**Figure 5. (upper panels)** Composite maps of Z500 anomalies [m] during 3-13 February 2018 for the 15 members with the lowest values of u60_10 minus the 15 members with the highest values of u60_10 after the onset date of the SSW in ERA5 (12-16 February 2018). Dots indicate statistically significant values at the 95% confidence level (t-test). Black boxes delimit the area of the tropospheric precursors of the SSW. (Bottom panel) Z500 anomalies in ERA5 for the same time period (3-13 February 2018). The magenta solid (green dashed) contours show the ERA5 climatological WN2 component of Z500 at ± 80 m.





**Figure 6. Same as Figure 5 but for Z500 anomalies during 18-30 December 2018. The dates considered for selecting the members with the strongest u60_10 and weakest u60_10 are 2-6 January 2019. The magenta solid (green dashed) contours show the ERA5 climatological WN1 component of Z500 at ± 80 m.**



**Figure 7. Same as Figure 5 but for Z500 anomalies during 1-10 September 2019. The dates considered for selecting the members with the strongest u60_10 and weakest u60_10 are 18-22 September 2019. The red solid (green dashed) contours show the ERA5 climatological WN1 component of Z500 at ± 40 m.**





In the ERA5 plot for SSW2018 (Fig. 5, bottom row), we identify three centers of action (Alaskan blocking, East North American trough, and Ural blocking), with the Alaskan blocking as the strongest one. This blocking and the East North American trough are located close to a positive and a negative antinode of the stationary WN2 component of Z500, which would lead to a constructive interference of anomalous and climatological WN2 waves (Nishii et al. 2009; Garfinkel et al. 2010; Rao et al, 2018). In the FREE experiments, the Z500 pattern associated with the weakest u60_10 displays, in general, similar features to ERA5. Notably, the "weakest u" members show stronger Alaskan blocking compared to the "strongest u" members in all systems except for GLOBO. This agrees with Martius et al (2009), who linked the occurrence of WN2 SSWs to blockings either over the Pacific or over both the Pacific and Eurasia. Nevertheless, in this case, the Atlantic and Eurasia centers also appear to be key for the occurrence of the SSW2018, consistent with Karpechko et al. (2018). The model that does not show this center of action in Fig. 5 (GRIMs) presents a very low number of ensemble members with an SSW.

For both SSWs in 2019, the Z500 patterns of ERA5 are characterized by a strong center of positive anomalies and slight negative anomalies out-of-phase by 180 degrees in longitude (Figs. 6 & 7). The pattern is consistent with the enhancement of WN1 wave activity leading up to these events previously shown. More specifically, for the NH event, the ERA5 pattern shows a blocking over Greenland with positive Z anomalies extending over the North Atlantic basin and Scandinavia (Rao et al. 2019) (Fig. 6 bottom row), whereas in the case of the SH SSW, a strong ridge is apparent close to the Antarctic Peninsula (Rao et al. 2020a, Shen et al. 2020) (Fig. 7 bottom row). In the FREE experiment for the SSWs of 2019, we also identify similar tropospheric patterns to ERA5 in both hemispheres associated with the occurrence of the weakest u60_10 relative to the strongest. In the NH SSW2019, the Greenland blocking preceding the SSW is mostly restricted to high latitudes, i.e., poleward of the stationary WN1 wave antinode, where constructive interference of waves would be less efficient. In the SH, the amplitude of the anomalies of Z500 prior to the SSW is very weak in most forecast systems. However, one must be careful with the interpretation of weak Z500 values for models in comparison with ERA5 in Figs. 5-7, as the models' plots are not showing a complete pattern of Z500 but just a comparison between two groups of ensemble members. These weak values of Z500 anomalies can be understood in two different ways: a high similarity (little spread) across ensemble members in both the troposphere and stratosphere, or a similarity only in the troposphere but with a different stratosphere– both of which could result in weak differences in the tropospheric patterns associated with the spread in the stratospheric forecasts. GLOBO for SSW2018 is an example of the former case, with u60_10 differing by only 8m/s between the two groups of ensemble members. In contrast, the behavior of most of the models during the SSW2019 SH fits the latter case, with very weak values of Z500 differences in Fig. 7 but differences in u60_10 ranging from 15 to 30 m/s depending on the model. Further analysis would be required to confirm these hypotheses.

To summarize the results of this Section, we aim to establish a connection between tropospheric anomalies, wave amplification in the stratosphere and ultimately, the occurrence of the SSW. To do so, we analyze the pairwise relationships among these variables, following the sequence of processes that link tropospheric disturbances to polar vortex weakening. First, we correlate the combined main centers of action of the modeled Z500 patterns of Figs. 5-7 with the stratospheric upward-propagating





wave activity, represented by the HF at 100hPa (HF100). For each SSW, we define the areas of these centers of action based on the extent of the main Z500 anomalies in all forecast systems and the locations of the antinodes of the climatological

WN1/WN2 waves in ERA5. The Z500 anomalies are then averaged over these regions denoted by black boxes in Figs. 5-7 and combined by computing the sum of averaged anomalies for centers with positive anomalies or positive-minus-negative when centers have opposite signs. The HF100 is computed for the predominant wavenumber and averaged during its strongest burst in ERA5 (6-14 Feb. 2018; 21Dec2018-3Jan2019; 2-15 Sep2019). As a second step, we assess the relationship between HF100 and the u60_10 around the SSW onset date.

Figure 8 presents scatter plots of the correlation coefficients corresponding to the pairwise relationships for each model and SSW event. These correlations are generally statistically significant, confirming the linear relationship among the three variables for all the SSW events analyzed in this study (see also Fig. S1). However, the strength of the relationship between tropospheric precursors and HF100 varies depending on the event. For SSW2018, the link between WN2 upward-propagating wave activity and tropospheric precursors is, in general, weaker compared to the other two SSWs. Moreover, there is a

substantial inter-model variability in the value of the correlation coefficient, ranging from 0.38 in GRIMs to 0.82 in UKMO-GloSea6 (Fig. 8a). In contrast, for SSW2019, WN1 wave activity in the lower stratosphere appears strongly correlated with the tropospheric precursors and models even agree on the value of the correlation coefficient (around 0.75) (Fig. 8b). Similar behavior is found for SSW2019 SH, but with weaker correlations and a broader spread across models (Fig. 8c). An exception is detected for the SSW2019 in CNRM-CM6-1, where HF100 and Z500 are decoupled (Fig. 8b). This model exhibits the

strongest HF100 (Fig. 3a), but the wave burst starts earlier than in ERA5 and the other models, and so, the timing of the wave amplification may explain the lack of coupling.

While the relationship between tropospheric precursors and HF100 varies across events, the correlation between the stratospheric variables remains approximately the same for the three events. u60_10 shows a strong negative correlation with the HF100, with correlation coefficient typically ranging from -0.55 to -0.85, although the coupling appears slightly weaker in

the SH than in the NH. Exceptions are found for SSW2018 in NAVGEM and the SSW2019 in CNRM-CM6-1. In the case of NAVGEM, several factors may contribute to the low correlation, potentially including biases in wave-mean flow interactions, at least in the NH, since this model also shows a relatively low correlation between u60_10 and HF100 during SSW2019 (Fig. 8a &b). For CNRM-CM6-1, as previously discussed, the timing of stratospheric processes is substantially earlier than in ERA5, with all ensemble members simulating an SSW around 25th December 2018.


These results indicate that a strong correlation between the two stratospheric variables does not necessarily imply a strong coupling between HF100 and the tropospheric precursors. In the case of the SSWs of 2019, the high correlation between HF100 and Z500 in all models suggests that tropospheric precursors highly influence the stratospheric wave activity, particularly prior to the NH SSW2019. Thus, inter-model differences in vortex deceleration may be related to variations in

wave-mean flow interactions as well as differences in the strength of tropospheric precursors. Conversely, for SSW2018, although there is a strong coupling between the stratospheric variables, the stratospheric wave activity might be modulated by



additional sources beyond the tropospheric circulation, at least in some of the models. This implies that the model's inability
to predict an SSW in this case may not only be due to an issue in representing tropospheric precursors, but also other
mechanisms such as non-linear wave processes in the stratosphere. Indeed, CNRM-CM6-1, the model with the largest number
of ensemble members with an SSW and the strongest burst of WN2 wave activity at 100hPa, does not show the highest
correlation between WN2 HF100 and Z500.

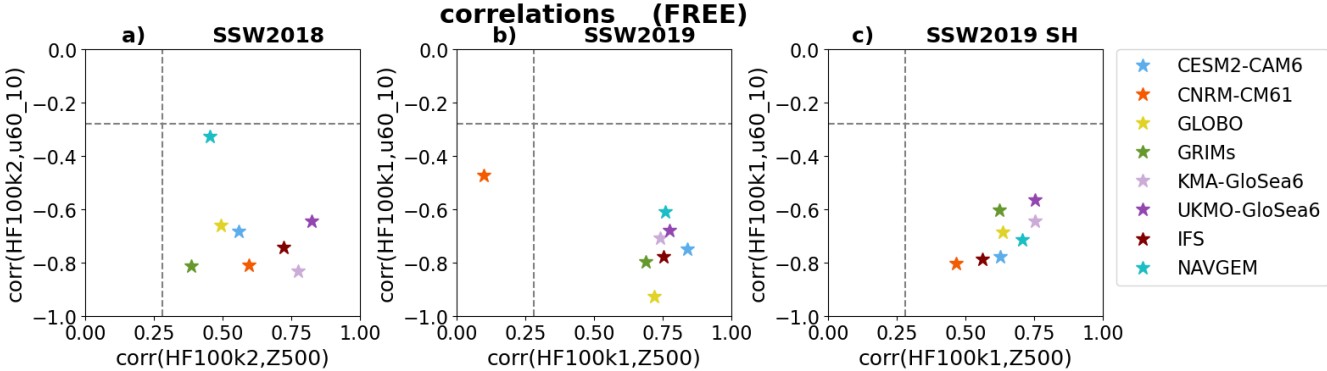

**Figure 8. Scatter plot of correlation coefficients across all members of each ensemble of u60_10 and HF at 100hPa for the**
**predominant wavenumber vs the correlation of this HF and tropospheric precursors identified in Fig. 5-7 at 500hPa (see more details**
**in the text). The gray dashed lines indicate the threshold values for statistically significant correlation coefficient values at the 95%**
**confidence level (t-test).**

## 4 Influence of the stratospheric state on triggering mechanisms of the three SSWs

So far, we have assessed model performance in capturing the triggering mechanisms of the SSWs in the FREE experiment and
hypothesized about the role of the troposphere/stratosphere in these mechanisms. Next, we investigate in more detail the
influence of the zonally symmetric stratospheric state on initiating the three SSWs by means of the NUDGED and CONTROL
experiments.

First, we analyze the stratospheric influence on the enhancement of upward-propagating wave activity prior to the SSWs.
Figure 9 shows, for each model, the ensemble distribution of total eddy heat flux at 100 and 300hPa averaged during the
strongest peak of HF at each level (indicated in the figure). Overall, the general deviations of the tropospheric and stratospheric
heat fluxes in the FREE experiments from the reanalysis are also shared by the NUDGED and CONTROL experiments. For
instance, the underestimate by the models of the enhancement of stratospheric wave activity prior to the SSW2018 in FREE is
evident in the NUDGED and CONTROL runs too. This might suggest a minimal influence of the stratospheric state on upward
wave propagation.
Nevertheless, when comparing results in the NUDGED and CONTROL experiments, some differences appear, although
mostly in the stratospheric wave activity. Prior to the SSW2018, the multimodel mean of HF100 in the NUDGED run is 44%



stronger than in the CONTROL run (Fig. 9a). Similarly, the HF100 preceding the SSW2019 SH is also stronger in the NUDGED than in the CONTROL experiment, but only by 17.8% (Fig. 9c). The sign of the HF100 difference agrees with linear wave theory for the SH event because a weak but still westerly polar vortex, as in the NUDGED experiment, favors upward wave propagation. For the NH SSW2019 case, the ensemble mean values of HF100 are very similar in the NUDGED and CONTROL experiments (the difference is only 8%) (Fig. 9b). The result points to a large tropospheric influence on the stratospheric wave activity leading to the occurrence of this SSW. This also confirms results of Fig. 8 for SSW2019 that showed the strongest correlation between the tropospheric precursors and HF100 for almost all models, with values around 0.75.

Interestingly, the dispersion of HF values across ensemble members in all models is very similar in the three experiments and the three events, suggesting that the stratospheric state does not modulate the variability of HF. The stratosphere does not seem to remarkably affect the tropospheric wave activity either, as we do not find relevant differences in HF at 300hPa across experiments.

The previous statements are based on the overall behavior observed across all models. However, although the zonal mean state in the sensitivity experiments is the same in all models, the performance of specific forecast systems varies. Notably, the CNRM-CM6-1 system stands out for the SSW2018 (Fig. 9a). Unlike the other models, the median value of the HF100 in the NUDGED experiment of CNRM-CM6-1 is substantially stronger than in the CONTROL one, with no overlap in their interquartile ranges. More importantly, only CNRM-CM6-1 shows an impact of the stratospheric state on the tropospheric wave activity, as the NUDGED median exceeds the CONTROL interquartile range of HF at 300hPa. Further, CNRM-CM6-1 shows the strongest HF values at 100 and 300hPa in both the NUDGED and FREE experiments and consistently, the highest number of ensemble members predicting an SSW (62%) in FREE. These results suggest that in a system that is more skillfully able to predict the factors that lead to this SSW, the stratospheric state does indeed have an influence on the troposphere. Further evidence will be presented next that indicates that the stratospheric state is having an influence on the tropospheric circulation patterns that may have played a relevant role in triggering the SSW2018.





**Figure 9. Boxplots showing the ensemble distributions of eddy heat flux (HF) at 100 and 300hPa and 45º-75º for the period with the strongest value of HF preceding each SSW for all models and the different experiments (FREE, NUDGED, and CONTROL runs).**



**The interquartile range (IQR) is represented by the size of the box and the horizontal black line corresponds to the median value. Whiskers extend from the box to a distance of 1.5 times the IQR. Outliers (colored circles) are defined as points with values greater than 1.5 times the IQR from the ends of the box. ERA5 values are represented by horizontal magenta lines.**

Next, we explore the influence of the stratospheric state on the tropospheric structures that preceded the three SSWs. The
spatial patterns of Z500 across the three experiments are very similar, suggesting that the stratospheric state does not remarkably alter the tropospheric circulation prior to these three SSWs (Figs. S2, S3 and S4). These spatial patterns also closely resemble those in ERA5 (bottom panel of Figs. 5-7), except for SSW2018, where the WN2 wave pattern appears rotated relative to both ERA5 and the climatological WN2 wave patterns (Figs. S2, S3 and S4).

Considering the lack of influence of the stratosphere on the spatial pattern, we next study the amplitude of the Z500 anomalies,
focusing on the same regions identified in Figs. 5-7 as related to the SSW occurrence. Figure 10 shows the ensemble distribution of the combined Z500 anomalies over these precursor regions, used in Fig. 8. The results are consistent with the intensity of HF at 300hPa in Fig. 9. The Z500 anomaly amplitudes in FREE tend to be weaker than those in ERA5 except for the SSW2019 (Fig. 10b). In the case of SSW2018 the median value of these amplitudes is exceptionally weak in all systems but CNRM-CM6-1 which is close to ERA5 one (Fig. 10a). In contrast, the median values of Z500 anomaly amplitudes in
FREE prior to SSW2019 SH, although lower than ERA5, are closer to reanalysis than in SSW2018.

When comparing NUDGED and CONTROL experiments, the results confirm that the impact of the stratospheric state on the intensity of tropospheric circulation preceding SSWs is also small. Indeed, the two experiments show virtually identical results for each SSW in terms of median values and variability. As was the case for HF at 300hPa, CNRM-CM6-1 is the only system that displays large differences between the two experiments, but in this case for all three SSWs. Indeed, for the SSW2018 the
NUDGED experiment of this model simulates very similar anomalies to ERA5, whereas the CONTROL one resembles the weak values of other models such as KMA-GloSea6, IFS or CESM2-CAM6 (Fig. 10a). More importantly, the NUDGED ensemble shows a notable reduction in spread compared to CONTROL and FREE experiments (Fig. 10a). This means that the stratospheric state exerts control on the tropospheric circulation, which only this model is able to capture. This influence is also evident in the CNRM-CM6-1 FREE run as it shows stronger values than the other models and might be key for triggering
the SSW2018 event, as already suggested by the HF results. For the other two events, the Z500 anomalies in the CNRM-CM6-1 NUDGED are stronger than in CONTROL and similar to FREE. However, the ensemble spread is not very different across experiments and the model values differ from ERA5. Thus, the stratospheric control over the troposphere might not be as important for triggering these two SSWs as for SSW2018.





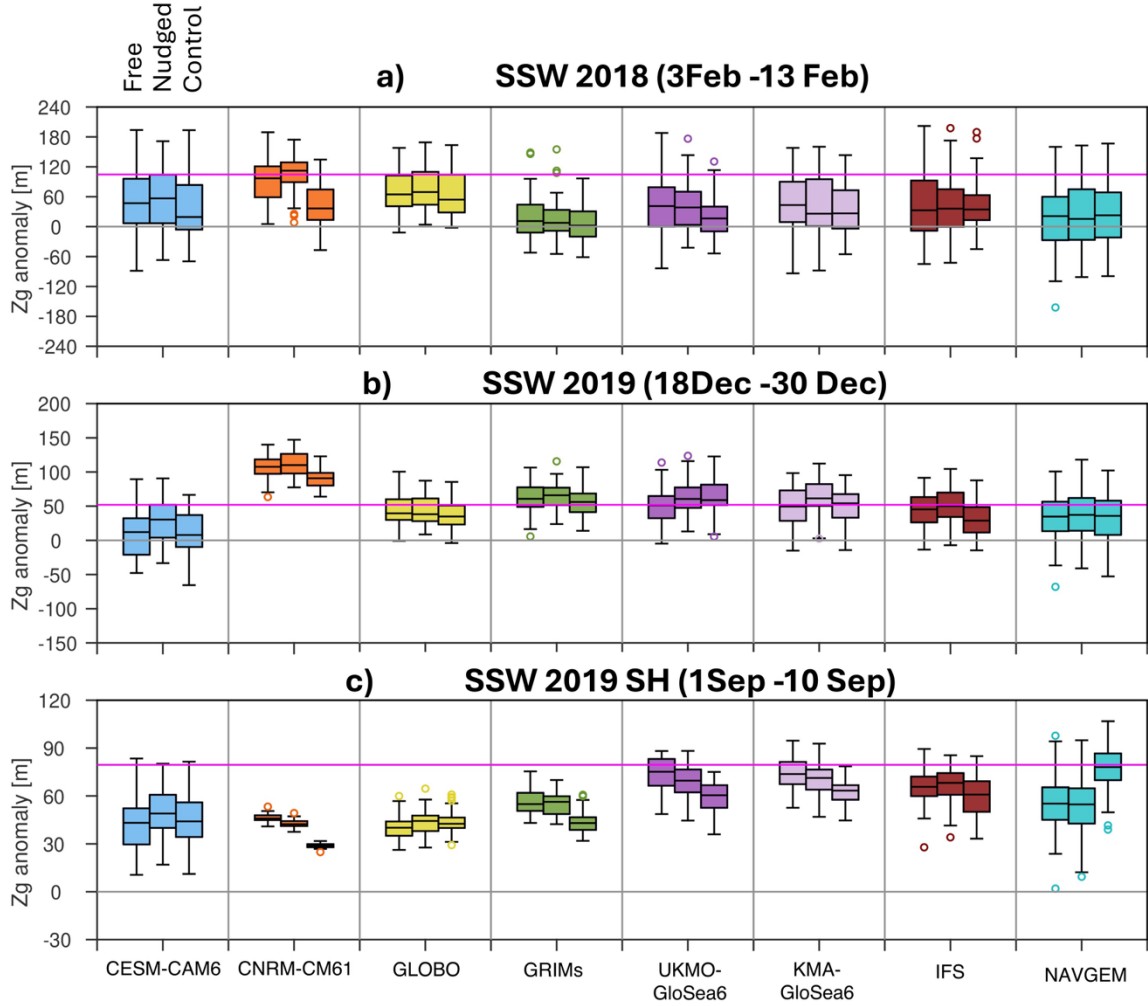

**Figure 10. Same as Fig. 9 but for the combined Z anomalies at 500hPa averaged over the areas of the tropospheric precursors of SSWs highlighted in Fig. 5-7. ERA5 values are represented by horizontal magenta lines.**

## 5 Study of the wave propagation and amplification prior to SSW2018

In the previous Sections, we have detected hints that the stratospheric state had the largest influence on the occurrence of SSW2018 (Fig. 9 and 10); a deeper analysis of this event with the SNAPSI experiments will thus help to clarify the role of the stratosphere in triggering this SSW. In addition, the SSW2018 is the most difficult event for the models to predict, so this analysis may also help to understand the models' issues with simulating the wave activity propagation. In the following, we perform a specific analysis of the wave propagation prior to the SSW2018.





## 5.1. Analysis of the upward wave propagation in the FREE experiment

We start by analyzing the vertical propagation of the wave activity in ERA5 and the FREE experiment of the models and the simultaneous evolution of the PNJ. Figure 11 displays the time evolution of the $F_z$ (shading) in both the troposphere and stratosphere for the predominant wave component (WN2). The zonal-mean zonal wind at 60ºN (u60) is also included in contours. For brevity, we only show the results for four models: CNRM-CM6-1, KMA-GloSea6, GRIMs, and IFS. CNRM-CM6-1 has the best forecast skill of this event; KMA-GloSea6 shows results that are close to those from UKMO-GloSea6;

GRIMs is included as an example of a low-top model since it performs similarly to the other two low-top models (CESM2-CAM6 and GLOBO). Lastly, IFS is an example of an intermediate model in terms of forecasting this SSW.

A first burst of wave activity is observed by late January in ERA5 and all models. The associated $F_z$ maximum is mainly restricted to the troposphere. In the stratosphere, weaker positive $F_z$ appears some days later, indicating that it propagates from the troposphere to the stratosphere, where it leads to a vortex weakening. After a period without upward wave propagation,

the polar vortex re-intensifies rapidly in the upper and middle stratosphere in early February (3rd-5th Feb) in ERA5 (Fig.11a) and shifts poleward with a PNJ centered at around 68ºN in that 3-day period (black contours in Fig. 12a). This is the time when the forecast systems start to deviate from the reanalysis. The strengthening of u60 is poorly simulated by the models (Fig.11b-e) and the polar vortex structure is also different, with a PNJ center shifted southward with respect to that in ERA5 (Fig. 12b-e). The PNJ configuration in ERA5 is characteristic of the "preconditioned state" of WN2 SSW events (Palmer, 1981; Albers

and Birner 2014).

As also shown in Fig. 2, ERA5 records a second, stronger burst of wave activity beginning on 7th February. This enhancement of wave activity occurs simultaneously in the troposphere and stratosphere. It decelerates the vortex and persists until the reversal of u60 on 12th February in most of the stratospheric column, when the wave activity starts to decline. Although the models also display high $F_z$ values in the troposphere, and to a lesser degree in the stratosphere, the wave enhancement is not

as intense and as simultaneous in the whole atmospheric column as that in ERA5, potentially explaining the absence of an SSW in most of the models and consistent with Fig. 3. Considering that the polar vortex state in the models starts to deviate from that in ERA5 at the same time or even at earlier stages than $F_z$, the stratospheric state might be important for the occurrence of the rapid enhancement of wave activity leading to the occurrence of the SSW2018.

## 5.2 Influence of the stratospheric state on wave propagation

To better understand how the stratospheric state affects wave propagation, we next analyze the refractive index in ERA5 and the FREE experiment. Figure 12 shows the Eliassen-Palm flux (F, arrows) and the refractive index squared ($n^2$, shading) for the WN2 wave component and the zonal-mean zonal wind (black contours) on 3rd-5th Feb 2018. This period corresponds to the time when the model deviations from ERA5 first become apparent. Note that for the models the probability of negative $n^2$ in the model ensemble is shown instead of $n^2$. This probability is defined as the fraction of members with negative $n^2$. Using

this metric avoids problems caused by overlapping positive and negative values when averaging different ensemble members.





Li et al. (2007) showed that this probability also provides a clear picture of the most probable atmospheric channels for wave propagation.

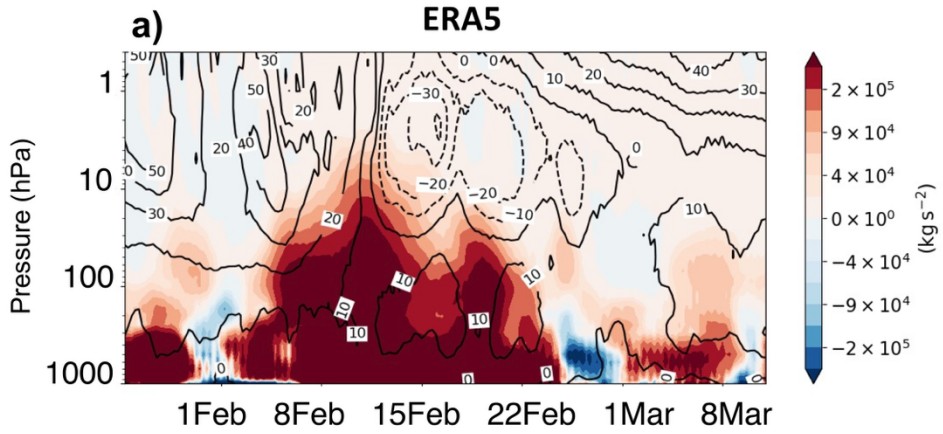

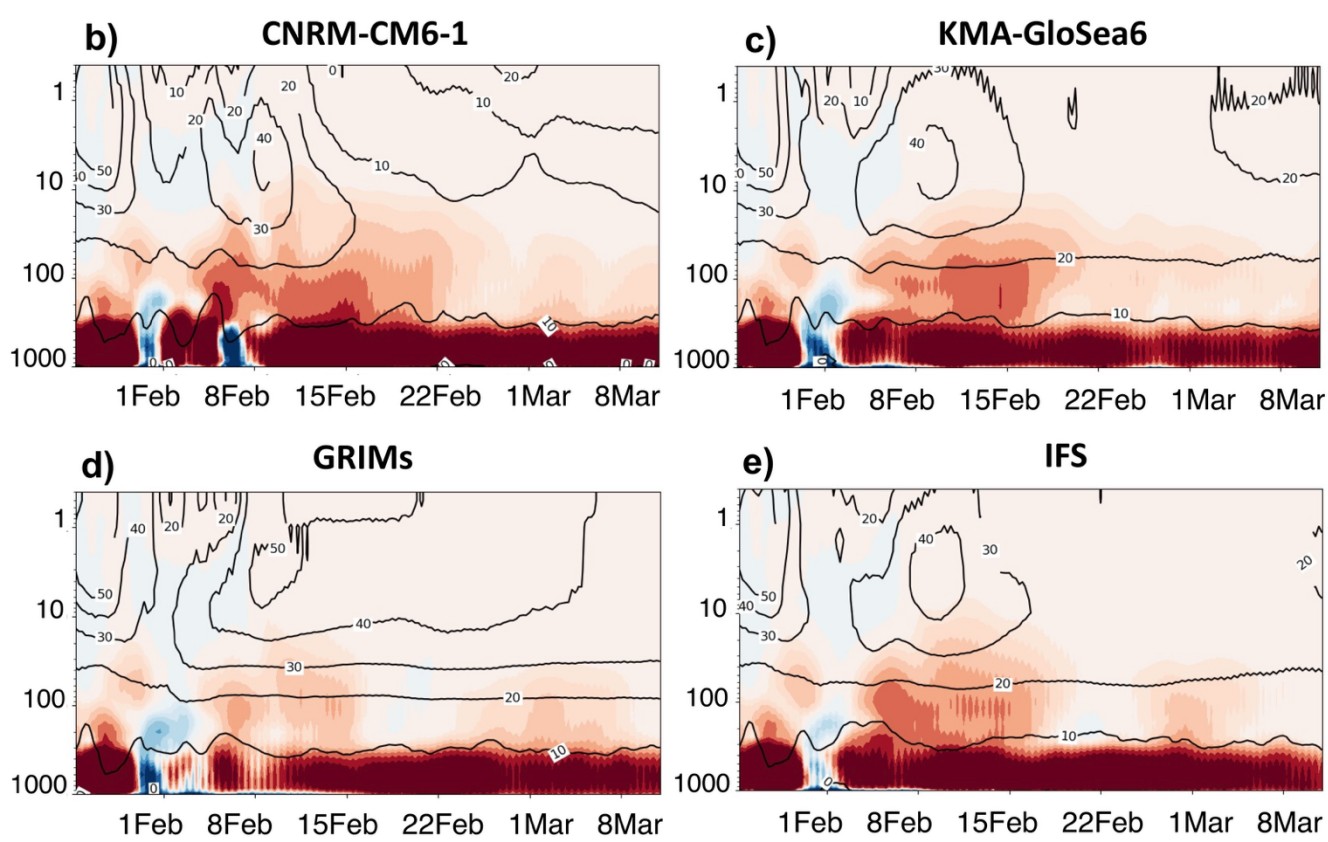

**Figure 11. Time evolution of the vertical component of the Eliassen-Palm flux ($F_z$, shading (kg s$^{-2}$)) averaged over 50º-70ºN for the WN2 component and the zonal mean zonal wind at 60ºN (black contours, m s$^{-1}$) from 25 January 2018 until 10 March 2018 in (a) ERA5 and the FREE experiment of (b) CNRM-CM6-1, (c) KMA-GloSea6, (d) GRIMs and (e) IFS.**



Consistent with Fig. 11, the period of 3rd-5th February is characterized by relatively weak wave activity that is simulated by reanalysis and models. Further, ERA5 displays a region around 50º-60ºN of relatively high values of positive $n^2$ that is approximately vertically aligned and extends from 100hPa up to 3hPa. The location of this region matches the edge of the PNJ and is thus consistent with a strong potential vorticity (PV) gradient (Fig. 12a). It might also provide a vertical waveguide. Between 3 and 1 hPa the extratropical region of positive $n^2$ shifts poleward to 55º-70ºN and so, negative values of $n^2$ appear around 45ºN-55ºN. This is consistent with the strong curvature of the wind associated with the region above the PNJ core and favors the poleward wave propagation in that area. Finally, there is also a band of negative $n^2$ horizontally aligned above 1 hPa and north of 55ºN. Overall, the spatial structure of $n^2$ and the poleward shift and vertical alignment of the PNJ resembles a wave cavity favoring the resonant excitation of waves in the mid-stratosphere and channeling waves towards the pole in the upper stratosphere that decelerate the PNJ at those levels (e.g., Smith 1989; Albers and Birner 2014). In contrast, when looking at the models' results, there is no similar structure in zonal wind or refractive index (Fig.12b-e). Indeed, models tend to show wider regions of low probability of negative $n^2$ between 10-3hPa than the corresponding area of high values of positive $n^2$ in ERA5. More importantly, in the upper stratosphere they do not simulate the poleward shift of the region of low probability of negative $n^2$. Consequently, a stronger equatorward wave propagation is displayed by models at those levels. Only CNRM-CM6-1 captures a small region of high probability of negative $n^2$ around 45º-50ºN between 3 and 1 hPa, which might explain the weaker equatorward component of EP flux with respect to the other models. However, waves are not channeled poleward in the upper stratosphere in this model either because they also find a band of high probability of negative $n^2$ between 55º-70ºN. The differences in $n^2$ are consistent with a different zonal-mean zonal wind structure where the PNJ is much weaker and shifted equatorward with respect to that in ERA5 (Fig.12b). The control of the upper-stratospheric wave propagation by the background state is confirmed by the analogous figures but for the NUDGED experiments, that do show the region of low probability of negative $n^2$ between 55º-70ºN (Fig. S5 & Fig. S6). Consistently, the equatorward component of EP flux vectors southward of 50ºN is reduced in NUDGED with respect to the FREE experiment, while waves in NUDGED are more likely turning towards the pole between 55º-70ºN than in FREE (Fig. S5 & Fig. S6).

The above atmospheric state undergoes substantial changes in the next 3-day period (6th-8th February) (Fig. 13). In ERA5 there is a marked intensification of upward-propagating wave activity in the stratosphere (channeled by positive $n^2$), the PNJ is weaker than before, and its maximum has descended to the middle stratosphere (Fig. 13a). The high positive values of $n^2$ observed in the stratosphere between 50º and 60ºN on 3rd-5th February shift equatorward in the next few days. The shift is particularly important in the upper stratosphere, where the previous dipole of $n^2$ between 45º-55ºN and 55-70ºN has disappeared and positive values of $n^2$ extend equatorward of 55ºN. This enhances the equatorward wave propagation. In addition, the horizontally aligned negative $n^2$ north of 50ºN has descended to about 5 hPa, an altitude that has been previously identified as favorable for efficient resonant internal wave excitation (Plumb 1981; Smith 1989). At the same time, in the lower and middle stratosphere the area with negative $n^2$ at high latitudes decreased, allowing upward wave propagation at higher latitudes (60º-72ºN), which in turn favors the PNJ weakening. The models also simulate an intensification of the stratospheric





wave activity (Fig. 13b-e), although weaker than in ERA5, particularly in GRIMs (Fig.13c). Specifically, equatorward wave flux in models tends to be notably weaker than that in ERA5, due to the different PNJ structure in the models in the mid- and upper stratosphere. The PNJ in models is stronger and larger than in ERA5 and this results in areas with high probability of negative $n^2$ between 30ºN and 50ºN, hindering the wave propagation in that direction. Additionally, the high probability of

negative $n^2$ at high latitudes above 10hPa is smaller in the models, making the equatorward wave propagation easier in models than in ERA5. Similar figures for the NUDGED experiment confirm the key role of the PNJ state on the wave propagation, in particular, on its meridional component (Fig. S7 & Fig. S8).

**Figure 12. (a) Eliassen-Palm flux (arrows, m³) and n² (shading, a⁻²) for the WN2 wave component and zonal mean zonal wind (black**
**contours) for 3rd-5th February 2018 in ERA5. (b)-(e) Same as (a) but for the FREE experiment of CNRM-CM6-1, GRIMs, KMA-GloSea6 and IFS. In (b)-(e) shading indicates the member fraction of negative n² instead of n².**





**Figure 13. Same as Fig. 12 but for 6th-8th February 2018.**

So far, we have seen that the wave propagation differs between the FREE model ensembles and ERA5 because of differences
in the background state. One may hypothesize that this could explain the weak wave activity in the stratosphere simulated by
the models. Next, we check if this is mainly due to the zonally symmetric stratospheric state by comparing the time evolution
of $F_z$ in the NUDGED and FREE experiments in Fig. 14, as the zonal mean stratospheric state in the NUDGED experiment
corresponds to the observations.

In general and similar to Fig. 9, in the days before the SSW we do not find markedly stronger upward-propagating wave
activity in the NUDGED experiment compared to FREE (Fig. 14), except in CNRM-CM6-1. The period of vortex
intensification on 3rd-5th February in NUDGED that is not captured by the FREE runs can be identified by the positive





NUDGED-minus-FREE differences in u60. After that, the FREE experiments do not capture the intensification of the wave activity in the stratosphere that was present in ERA5 from 6th February (Fig. 11). Interestingly, the NUDGED experiments of KMA-GloSea6 and GRIMs show even weaker wave activity in the first days of this period than the FREE ones. In CNRM-

CM6-1 and IFS, $F_z$ shows larger values in NUDGED than in FREE, although in the latter it is only statistically significant above 100hPa. More importantly, from 8th February none of the models show stronger wave activity in the troposphere in the NUDGED run than in the FREE one. Indeed, if they show a significant NUDGED-minus-FREE difference in those levels, it is negative, as seen in CNRM-CM6-1 and IFS (Fig. 14a & d). In contrast, in the stratosphere (in particular, from 50hPa) there is, in most of the models, an enhancement of upward-propagating wave activity. Notably, this enhancement is not connected

to a previous enhancement in the troposphere, i.e., the amplification occurs in the stratosphere itself, which constitutes another hint of wave resonance. These results suggest that the misrepresentation of the zonally symmetric stratospheric state in models contributes to the lack of the strong WN2 burst in the stratosphere. However, this contribution is smaller than the total model bias; namely, the models in the NUDGED experiment still simulate much weaker upward wave flux in the stratosphere than ERA5. Further, the mentioned wave resonance might be characteristic of the vortex breakdown and so, it is approximately

captured by the NUDGED experiment. However, it is not the cause of this breakdown, considering that the upward wave flux is not comparable to observations.

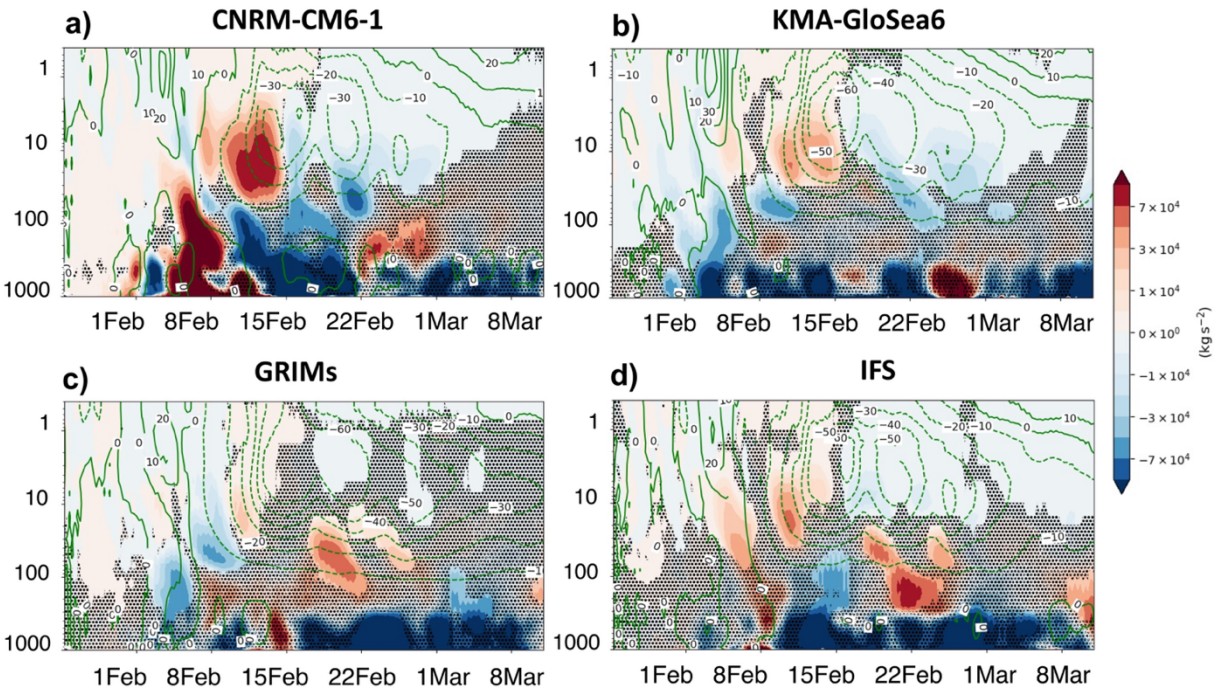

**Figure 14. Time evolution of the vertical component of the Eliassen-Palm flux ($F_z$, shading (kg s$^{-2}$)) averaged 50º-70ºN for the WN2 component and the zonal mean zonal wind at 60ºN (m s$^{-1}$) from 25 January 2018 until 10 March 2018 in the NUDGED-minus-FREE**

**experiments of (a) CNRM-CM6-1, (b) KMA-GloSea6, (c) GRIMs and (d) IFS. Dots indicate non-statistically significant values at the 95% confidence level (t-test). The vertical magenta line indicates the date of the SSW in ERA5.**



In the previous figures, the upward wave activity flux in the stratosphere has been found to be partially modulated by the zonal mean stratospheric state. As a final step, we assess the influence of this state on the net EP flux budget in the polar stratosphere during the strongest peak of wave activity in the stratosphere (6th-14th February) (Fig. 15, see Section 2.2 for calculation details). Figure 15 presents the ensemble distribution of the net EP flux for all wavenumbers (Fig. 15a) and WN2 only (Fig. 15b) in the polar stratosphere (55º-90ºN and 100-10hPa) along with the different terms of the latter, i.e., the integrated $F_z$ at 100 hPa ($F_{100}$) and 10hPa ($F_{10}$) and the $F_y$ at 55ºN ($F_{55N}$) (Fig. 15c-e). The positive values of the net EP flux in ERA5 indicate strong EP flux convergence for all wavenumbers (horizontal black line in Fig. 15a) and for WN2 only (Fig. 15b) in the stratosphere, consistent with a strong deceleration of the PNJ. There is a remarkable upward WN2 EP flux at 100hPa in agreement with Figs. 2, 9, and 11 (horizontal black line in Fig.15c). Conversely, the upward flux at 10hPa ($F_{10}$) and the equatorward EP flux at 55ºN ($F_{55N}$) display much weaker values (Fig.15d & e, respectively). Thus, the $F_{100}$ constitutes the main contribution to the strong convergence of WN2 wave activity in the polar stratosphere and, ultimately, to the occurrence of the SSW2018. However, in the FREE experiment, models show the largest issues with simulating this term, showing much smaller values than ERA5 (Fig.15c), which leads to relatively low positive values of net EP flux for WN2 (Fig. 15b) and thus almost no impact on the PNJ strength.

Focusing now on the effects of the stratospheric state on the EP flux budget, we do not find large differences in the magnitude of the net budget or in any individual term across the sensitivity experiments. The only exception is $F_{55N}$, which appears to be modulated by the stratospheric state, as all models show higher $F_{55N}$ values in the NUDGED than in the CONTROL runs. However, this difference is statistically significant only in CNRM-CM6-1 (Fig. 15e). This result implies that the zonal mean stratospheric state influences wave propagation by enhancing the equatorward EP flux. The increase of this equatorward flux has already been detected in the 6th-8th February (Fig. 13) and was found to be influenced by the observed weaker vortex (Fig. S7). This modulation of $F_{55N}$ impacts the net EP flux budget, with lower values in the NUDGED than in the CONTROL experiments, especially in CNRM-CM6-1. Another important remark is the impact of the stratospheric state on the spread of the net EP flux and its components: In CNRM-CM6-1 and UKMO-GloSea6, the spread of all the analyzed quantities is narrower in the CONTROL than in the NUDGED experiments. This likely occurs because the stronger vortex in CONTROL imposes greater constraints on wave propagation.



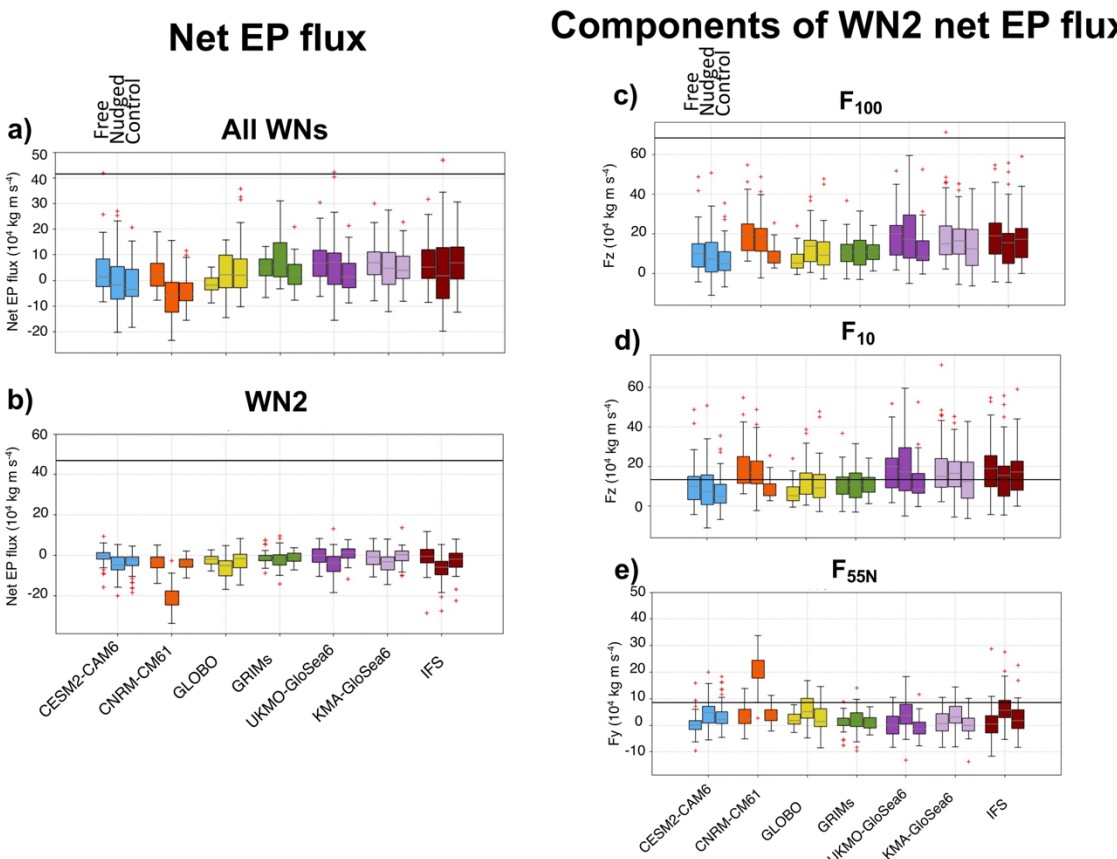

**Figure 15.** Boxplots of net EP flux in the polar stratosphere (100-10hPa, 55º-90ºN) for all wavenumbers (a) and WN2 (b) and its components (c)-(e) averaged during 6th-14th February 2018. The interquartile range (IQR) is represented by the size of the box and the horizontal black line corresponds to the median value. Whiskers extend from the box to a distance of 1.5 times the IQR. ERA5 values are represented by horizontal black lines. The outliers are indicated by red crosses.

## 6. Summary and discussion

In this study, we perform a multimodel analysis of triggering mechanisms for SSWs depicted by S2S forecast systems. In particular, we take advantage of the new SNAPSI experiments from eight S2S forecast systems to systematically assess, for the first time, the role of the stratospheric state in triggering the boreal SSWs of 2018 (SSW2018) and 2019 (SSW2019) and the austral minor warming of 2019 (SSW2019 SH). The SNAPSI experiments represent a unique set of experiments specifically designed to isolate the effects of the zonally symmetric stratospheric state on the atmosphere, with the stratospheric state either evolving freely, following the climatology, or following the specific SSW conditions depending on the experiment.

These are the main conclusions from the analysis of the SNAPSI experiments:




- Forecast systems show a similar quality for predicting the upward wave activity flux in the troposphere and stratosphere prior to the three SSWs as for predicting the events themselves, in agreement with Taguchi (2018). Specifically, while the forecast systems are able to simulate the moderate but persistent WN1 wave activity preceding both SSW events of 2019 (SSW2019 and SSW2019 SH), they fail to capture the intense burst of WN2 wave activity that occurred before the SSW2018. Similarly, the models have the most difficulty in simulating the SSW2018, while their performance is relatively good for the other two SSWs.

- The forecast systems perform better at simulating the tropospheric circulation before the two SSWs of 2019 than before the SSW2018. In particular, the models capture the location (near the antinodes of the climatological WN1 Z500 wave) and intensity of the blocking highs preceding the two SSWs of 2019. In contrast, tropospheric precursors prior to SSW2018 in models exhibit weaker intensity than in ERA5 and are not even simulated in many ensemble members. This explains why the models could simulate the enhancement of wave activity in the troposphere through constructive interference of anomalous and climatological waves in the case of the 2019 events, while failing to predict this enhanced tropospheric wave activity for the 2018 case.

- The zonal mean stratospheric state does not seem to drastically influence the variability and mean values of the upward wave activity flux and tropospheric circulation anomalies prior to SSWs. The largest influence is identified for the stratospheric wave activity, although this modulation depends on the event characteristics. The relative role of the stratospheric state in triggering the SSW also varies by event. For instance, the SSW2019 appears to be primarily driven by tropospheric processes, as the stratospheric influence on the extratropical upward-propagating wave activity at 100hPa and on the tropospheric precursors is small while the intensity of these precursors is highly correlated with the stratospheric wave activity at 100hPa in most of the models. In contrast, the contributions of the stratosphere may have played an important role in triggering SSW2018 and SSW2019 SH.

- The SSW2018 was preceded by a very strong burst of WN2 wave activity that occurred simultaneously in the troposphere and stratosphere. The poleward-shifted polar vortex state may have conditioned the occurrence of this strong burst, but the models were not able to capture it in the FREE experiment. Even when the zonally symmetric stratospheric state is imposed, the forecast skill of this burst is still low, indicating that other factors such as the tropospheric precursors are fundamental for the strong enhancement of WN2 in the whole column.

Overall, the results suggest that both tropospheric and stratospheric processes play a role in initiating an SSW, although their relative contributions vary by event. As just mentioned, the SSW2019 represents a clear example of an event mainly driven by tropospheric precursors in agreement with Matsuno (1971). The polar vortex was initially close to a climatological state and gradually decelerated as WN1 wave activity slowly amplified. This finding is then consistent with theories that suggest that WN1 events are typically related to tropospheric processes (Quiroz, 1975).

The austral SSW2019 SH was also preceded by WN1 wave activity, but in this case the stratospheric state played an important role in the vortex disruption. The climatological SH polar vortex is typically strong enough to prohibit most of the wave



propagation. In the 2019 winter it was initially weaker than usual, probably due to the combined effect of an easterly quasi-biennial oscillation phase and solar minimum (Rao et al. 2020a, Shen et al. 2020). We show that this initially weak polar vortex was key for the enhancement of wave activity in the stratosphere, though an anomalous flux of WN1 wave activity from the troposphere was also a necessary factor.

Both the stratosphere and troposphere were also key for the occurrence of the WN2 SSW2018 in the NH. The PNJ was initially stronger than usual. After a wave burst propagating from the troposphere to the stratosphere, the PNJ shifted poleward, leading to what has been called as a "preconditioned state" for WN2 SSWs (e.g., Palmer, 1981; Charlton and Polvani, 2007; Albers and Birner, 2014). Thus, we see that tropospheric processes are important for, at least, tuning the vortex into this state. However, the second and strongest burst of wave activity preceding the SSW did not propagate vertically (indicated by a lack

of vertical phase tilt, e.g., Salby, 1984; Albers and Birner, 2014). This is characteristic of wave resonance in the stratosphere that is also linked to the existence of this "preconditioned vortex". Consistently, stratospheric wave activity is enhanced in the NUDGED experiment where the observed zonal mean stratospheric state is imposed, however wave activity is still much weaker than in ERA5.

The model with a high number of ensemble members with an SSW in the FREE experiment (CNRM-CM6-1) is the only one

that shows a significant influence of the stratospheric state on the wave activity in the lower stratosphere and, more importantly, on the intensity of tropospheric precursors. This is consistent with recent studies (e.g.: Albers and Birner 2014; de la Cámara et al. 2017) that show that the strong resonance-induced wave burst that accompanies the SSW at all levels would be one of the features of the vortex breakdown but not the cause. Thus, we can hypothesize that only models that capture wave resonant processes, particularly those predominantly driven by the stratospheric state, are able to forecast this type of event. However,

we found that the magnitude of the net EP flux in the polar stratosphere, as well as the differences between the models and ERA5, is largely determined by the vertical component of the EP flux entering that region (100 hPa). Since this magnitude is commonly considered to measure the injection of tropospheric wave activity into the stratosphere (e.g., Hu and Tung, 2003), one might argue that the net EP flux in the stratosphere is dominated by tropospheric precursors. Furthermore, while the stratospheric wave activity is enhanced in the NUDGED experiment, it is dramatically different from other experiments during

the strong wave burst only in the CNRM-CM6-1 model.

Several factors may explain these apparent inconsistencies: First, recent studies (e.g., de la Cámara et al. 2017) have demonstrated that the extratropical eddy heat flux at 100hPa is already modulated by the stratospheric state. Indeed, in the case of SSW2018, the stratospheric wave activity appeared to be modulated by additional sources beyond the tropospheric circulation as the correlation between the tropospheric precursors and the WN2 stratospheric wave activity in more than half

of the models, although significant, was weaker than in the other two SSWs. Second, if resonance is at play, $F_z$ pointing upwards does not mean that the source of wave activity is at lower levels. This interpretation is only valid under linear propagation conditions, but in the context of resonance growth is likely non-linear. Indeed, the relevance of non-linearity wave propagation has been also suggested by the relatively lower linear correlation between tropospheric circulation and HF100. Moreover, unlike in the other two SSWs, there is a large disparity in the values of this correlation across models, suggesting



that these processes might be model dependent or event that only some models can capture them. Finally, it is important to note that in the NUDGED experiment, only the zonally symmetric stratospheric state is prescribed, but the asymmetric part is different in each model. In the days before a WN2 SSW, the vortex is highly nonzonal and zonal asymmetries may significantly influence wave propagation. Indeed, the zonally asymmetric gravity wave drag has been identified as important for triggering wave resonance (Holton 1984), and gravity wave drag remains a source of uncertainty in models.


Using the SNAPSI experiments also enabled comparison of the ability of the forecast systems to simulate processes involved in the occurrence of SSWs. Previous studies (e.g., Taguchi 2018; Domeisen et al. 2020a; Chwat et al., 2022) have shown that S2S forecast systems have more difficulty forecasting split than displacement SSWs. A similar bias has been documented in climate models, which underrepresent split SSWs (Hall et al. 2021). From the present work, we can confirm those results for

the three events of this study. This may seem surprising since a very recent study has shown that the upward coupling of WN2 wave activity between 500 and 100 hPa in S2S forecast systems is better captured than that for WN1 (Garfinkel et al. 2025). However, we must be cautious in this comparison, because while Garfinkel et al. (2025) performed a climatological analysis, we are focusing here on extreme events. Indeed, one of the advantages of our analysis relies on the evaluation of the ability of the forecast systems to simulate variations in processes involved in triggering different SSW events.

Nevertheless, this study does have some limitations. First, the SNAPSI experiments have only two initializations for each SSW and only one prior to the occurrence of the event. This prevents us from evaluating when the forecast skill of certain processes improves. Second, all conclusions are drawn from the analysis of only three SSW events so we must be cautious with any generalizations. Future work could further capitalize on the SNAPSI protocol by extending these experiments to more events and including different types of stratospheric variability.

To conclude, the SNAPSI experiments using different S2S forecast systems have allowed us, for the first time, to systematically identify the relative roles of the troposphere and stratosphere in triggering SSWs. By prescribing different stratospheric states, we were able to isolate their effects on this triggering during the SSWs. This is particularly relevant because in the few days before the SSWs, the triggering mechanisms and their effects are already active, making it difficult to disentangle cause and effect. The SNAPSI framework complements other methodologies used to analyze SSW triggering mechanisms, such as

experiments where the tropospheric variability is suppressed (Scott and Polvani, 2004) or prescribed (de la Cámara et al. 2017). Indeed, although this kind of nudging is not new, as far as we are aware it is the first time that it has been applied to study SSW onset. In the future, a similar initiative testing the influence of other factors in the occurrence of SSWs, such as parameterization of gravity waves (McLandress et al. 2013; Polichtchouk et al. 2018) or interactive chemistry (Oehrlein et al. 2020), will help further advance understanding of the occurrence of SSWs.


*Data availability.* The ERA5 data used in this study comes from https://cds.climate.copernicus.eu/datasets.

The          SNAPSI          forecasts          used          in          this          study          are          archived          by          CEDA
https://catalogue.ceda.ac.uk/uuid/0a5a1ce22fb047749e040879efa8e9b5.



*Supplement.* There is an available supplement related to this article.

*Author contributions.* B.A., A.H.B., P.H. and C. I.G. designed the study. B.A. wrote the first draft of the manuscript. Z.D.L., B.A., W.N., P.R., D.D.M. and Z.W. performed the analysis. P.H., A.H.B., and C.I.G. coordinated the SNAPSI runs. C.B, D.-C.H., Y.-K.H., H.K., J.K., P.M., D.M., J.O, I.P., J.H.R., S.-W.S, D. S., I.R.S. and T.S. ran the individual model and contributed
the output. All authors discussed the results and edited the manuscript.

*Competing interests.* At least one co-author is member of the editorial board of Weather and Climate Dynamics.

*Acknowledgments* B.A. and N.C. acknowledge support from the project Stratospheric Ozone recovery in the Northern
Hemisphere under climate change (RecO3very): PID2021- 124772OB-I00 from the Spanish Ministry of Science and Innovation. SWSON was supported by the Institute of Information & communications Technology Planning & evaluation (IITP) grant funded by the Korea government (MSIT) [NO.RS-2021-II211343, Artificial Intelligence Graduate School Program (Seoul National University)]. IRS and JHR acknowledge funding from the NSF National Center for Atmospheric Research, which is a major facility sponsored by the NSF under Cooperative Agreement No. 1852977. AdlC was supported
by the grant PID2022-136316NB-I00 funded by the Spanish Research Agency. GLM was supported by the Jet Propulsion Laboratory (JPL) Microwave Limb Sounder team under JPL subcontract #1521127 to NWRA. C.I.G is supported by the ISF-NSFC joint research program (ISF grant No. 3065/23) and the NSF-BSF joint research program (United States-Israel Binational Science Foundation grant no. 2021714). M.O. is supported by the Agencia Nacional de Promoción Científica y Tecnológica (grant no. PICT-2021-GRF-TI-00498) and the Universidad de Buenos Aires (grant no. 20020220100075BA).
Z.W is supported by the University at Albany - State University of New York (grant no. 940001-65). This work used JASMIN, the UK's collaborative data analysis environment (https://www.jasmin.ac.uk) (Lawrence et al 2013)

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
