# Peer review of "The role of the stratospheric state in upward wave flux prior to Sudden Stratospheric Warmings: a SNAPSI analysis"

_EGUsphere, 2025_

## Referee Comment (RC2)

**Review of 'The role of the stratospheric state in upward wave flux prior to Sudden Stratospheric Warmings: a SNAPSI analysis' by Ayarzagüena et al.**

**General comments:**

This study analyzed how the stratospheric condition influences the upward wave flux, and thus the occurrence of SSWs, based on a set of S2S model experiments in SNAPSI. The S2S forecasts are still struggling to predict SSW with long lead times. Thus, the current results provide helpful insights into understanding the role of stratosphere, which could potentially help improve the forecast skill in the future. The manuscript is very clean and well-written. However, since there are various diagnostic metrics and a huge amount of information, a major revision is needed to help with the presentation and delivery of the main conclusions.

**Major Comments:**

1. Balance in presenting the model mean and model spread. Currently, most analyses are based on the model mean, while the model spread is not fully presented. For example, in Section 3, while Section 3.1 shows the mean skill of models in predicting zonal wind, Section 3.3 focuses on the difference in Z500 between 'weaker u' and 'strong u' groups. This 'inconsistency' poses a mismatch in the obtained information. It would be helpful to link the spread in Z500 with the spread in zonal wind. Thus, the model spread in predicting the zonal wind can be added in Fig. 1, and the model mean of Z500 should be included in Figs. 5-7. Although the fraction of members forecasting an SSW is shown in Table 3, this cannot accurately reflect the spread. For the latter, including the model mean will help in interpreting Figs. 5-7 as discussed in L360-365.

2. Discussion on the stratospheric state. While I understand that the main focus of this study is to provide a general understanding of the role of the stratospheric state, it would be worthwhile to add more discussion on what might constitute the stratospheric state. For instance, for the 2018SSW, the poleward shift of the PNJ is missing in the models, which is one possible candidate. In addition, the manuscript did not discuss about the QBO (although very briefly in L686), which is also included in the stratospheric condition and reflected by the experiment design. Although at such a short timescale the bias in the QBO may not be evident, the three SSWs occurred under different QBO states (e.g., Butler et al. 2020; Shen et al. 2020), and it would be worthwhile to reflect this and discuss the potential implications. In addition, as briefly discussed in Section 3.1, the intensity of the polar vortex relative to climatology also differs, which could also serve as preconditioning. The related discussion should be added to provide general implications.

Reference:

Butler AH, Lawrence ZD, Lee SH, Lillo SP, Long CS. Differences between the 2018 and 2019 stratospheric polar vortex split events. Q J R Meteorol Soc. 2020; 146: 3503–3521. https://doi.org/10.1002/qj.3858

Shen, X., Wang, L., & Osprey, S. (2020). Tropospheric forcing of the 2019 Antarctic sudden stratospheric warming. Geophysical Research Letters, 47, e2020GL089343. https://doi.org/10.1029/2020GL089343

3. Visualization of the plots. There are quite a few plots which are all informative. However, for some plots it is difficult to identify the regions/features being discussed in the main text. I suggest the authors adjust the plots to make the information more straightforward, which will help readers grasp the key information more quickly. Please see the detailed comments in the Specific Comments.

**Specific Comments:**

1. L210: Suggest explaining that the negative eddy heat flux indicates upward propagation in the SH. Despite the figure caption of Fig. 4, there is no explanation in the main text. Please add this.

2. L227: The classification of 'weakest u' and 'strongest u' is a bit counterintuitive, as the 'weaker' group corresponds to a better forecast, whereas the 'stronger' group corresponds to a worse forecast. Would it be better to define them as something like 'SSW-like' and 'no SSW', which is more straightforward?

3. L260, Figure 1: Suggest also showing the spread in predicted [u] as in my major comment 1.

4. L278: Suggest briefly stating that WN2 is mainly responsible for the 2018 SSW, otherwise it is a bit abrupt to focus on WN2 directly.

5. Figures 2-4: Since there are lots of lines, I suggest bolding the model being discussed in the main text for visualisation.

6. Figures 5-7:

- Suggest adding the variance of forecasted [u] among the members in the subtitles after the model's name. This could help provide a straightforward comparison and understanding of the linkage between the spread in tropospheric circulation and stratospheric response.
- Suggest also adding the box to indicate the region in ERA5, as the map is not very visual and thus takes time to identify the region of focus.
- The climatological PWs are from ERA5. While I understand this is what one can do, it is also possible that the model bias in the climatological PWs can influence the interpretation of linear interference. Perhaps it would be better to include a brief discussion.

7. L348: Please clarify what 'this center of action' refers to.

8. L380-410: Suggest checking the HF300 as well. As stratospheric wave forcing does not entirely originate from tropospheric forcing as discussed later on and shown in Yessimbet et al. (2022). It would be helpful to establish a linkage between the tropospheric circulation, the tropospheric wave forcing (HF300), and the stratospheric wave forcing (HF100).

Reference:

Yessimbet, K., Shepherd, T. G., Ossó, A. C., & Steiner, A. K. (2022). Pathways of influence between Northern Hemisphere blocking and stratospheric polar vortex variability. Geophysical Research Letters, 49, e2022GL100895. https://doi.org/10.1029/2022GL100895

9. Figures 8 and S1. I like Fig. S1, which is very informative. It not only confirms the linear relationships among the three variables but also indicates the spread among the model members. I suggest the authors move it to the main manuscript, perhaps merging it with Fig. 8 as they are related, and add more discussion on it. For instance, we can see that for SSW2018, the scatters are densely located in the upper left, indicating that the forecast [u] is overall quite strong and related to the weak HF (Fig. S1a). Moreover, for the HF and Z500, the scatters are located in the lower panel (Fig. S1d). Despite the weaker Z500-HF linkage, the Z500 is also quite diverse. For comparison, it would be better to use the same range for x and y axes across different cases, also add the corresponding ERA5 variables. This will also help to interpret the conclusion from Fig. 8.

10. L432-437. Please briefly explain how the quantitative changes are computed.

11. L440: It is interesting that for 2019SSW SH, although the models have relatively good performance in capturing the tropospheric wave forcing (i.e., for the multiEnsM, the observation is within the 1.5IQR), they fail to capture the observed stratospheric wave forcing. This seems to imply that the stratospheric wave forcing does not completely come from the tropospheric wave forcing. Whereas in the other two events, the skill for HF300 and HF100 is more similar.

12. L477: Suggest adding 'for the majority of models' for clarity.

13. L480: Suggest changing 'model simulates very similar anomalies to ERA5' to 'the ERA5 value lies within the IQR ..'

14. Figure 11: Suggest marking the period of interest.

15. L537-538: Suggest changing 'relatively weak wave activity that is simulated by reanalysis and models' to '...wave activity seen in the reanalysis and simulated ...'

16. Figures 12-13: Suggest adding the box to indicate the regions for EP flux budget and changing the y-axis labels to hPa for consistency with the main text. In addition, suggest extending the latitudes to 10S to show the QBO structure, related to my major comment 2.

17. Figure 14: Please make sure the vertical magenta line mentioned in the figure caption is visible.

18. Figure 15: Suggest changing the color of the median value line for visualization. In addition, while currently we can see the difference in Fz and Fy, we cannot find their relative contribution to the net EP flux convergence. According to the previous analysis (Figs. 12, 13), the difference in the horizontal wave forcing is evident, however, it does not appear to play an important role according to this plot. This poses a gap between these two sections. I suggest the authors add a scatter plot of Fz100 vs Fy for each member, similar to Fig. S1. This might help in understanding their relative roles.

---

## Author Comment (AC1)

**REVIEWER 1**

Ayarzagüena et al. investigate how the stratospheric background state influences upward wave activity preceding Sudden Stratospheric Warmings (SSWs). Using ensembles of different models from the Stratospheric Nudging And Predictable Surface Impacts (SNAPSI) project, the study compares free-running, stratosphere-nudged and control simulations to ERA5 for three events: the February 2018 boreal SSW, the January 2019 boreal SSW, and the September 2019 austral minor SSW. The study aims to isolate the effect of the stratospheric state on the triggers of SSWs. Overall the paper provides a detailed study of the processes and wave fluxes that go into contributing to the different SSWs and attempts to separate tropospheric and stratospheric influences. The paper itself is rather long but the analysis is thorough and the authors walk the reader through the plots and their interpretation. I would recommend publication after addressing the points below.

Thanks a lot for your comments. Here is our reply to the major and minor comments in blue:

1) Figure 1 and Table 3. I have difficulty reconciling the fact that the ensemble mean line for CNRM in Fig 1(a) doesn't show an SSW but 62% of the ensemble members do. Do you have a suitable plot to illustrate the spread please?

First, it is important to note that we calculate the percentage of ensemble members predicting an SSW event during the 45 days of the FREE experiment. Therefore, it is not restricted to a specific day or a short period around the observed SSW date. In the CNRM-CM6-1 ensemble, SSW events occur between 8th February and 11th March 2018 (the final day of the simulation), with a higher probability between 15th and 28th February (Figure R1.1). This temporal spread may then partially explain why the ensemble mean of u60_10 does not cross the zero-line on any of the 45 days of the FREE experiment, but it falls below the ERA5 climatological values from 14th February onward as shown in Figure 1a of the manuscript.

[Figure]

**Figure R1.1.** *Distribution of SSW events in the CNRM-CM6-1 ensemble members during the 45-days period simulated in the FREE experiment for the SSW2018.*

Following Reviewers 1 and 2 recommendations, we have looked for a systematic way of displaying the ensemble spread of u60_10 for all models. Doing this for each day and model in Figure 1 may not be straightforward, given that we include eight models plus the two ERA5 lines. However, we agree on the importance of illustrating the model spread for at least a

relevant period, such as the dates surrounding each SSW in ERA5 (12th-16th February 2018, 2nd-6th January 2019 and 18th-22nd September, Table 2 of the manuscript). To address this, we have added three new panels to Figure 1 (Fig.1d-f) that show the ensemble distribution of u60_10 for these periods. A brief discussion of the ensemble spread in each case has also been included in the revised manuscript (L251-260 of the revised manuscript). For instance, these panels reveal that easterly wind values fall within the 1.5 interquartile range of the ensemble distribution for the CNRM-CM6-1 model during the SSW2018, whereas this is not the case for the other models. This helps then reconcile the wind values and the % of ensemble members simulating an SSW.

2) Figure 5 to 7: I think all the subplots should use the same colorbar within each figure. For ERA5, am I right in thinking that this is a difference from climatology whilst for the models it is a difference between the strongest and weakest ensemble members? As such I would expect that the model composite differences shown are larger than if you were able to do a comparison to the free running climatology of each model (like for ERA5). I would like to see a more careful discussion of what is being shown in these figures around line 320. Whilst there are similarities between the patterns in some models and ERA5, the strength is much weaker in all cases.

Figures 5 - 7 display the difference of the anomalous Z500 for the "weakest u" members (the 15 members with the weakest polar night jet (PNJ) at the time the SSW was observed in ERA5) minus the anomalous Z500 for the "strongest u" members (the 15 members with the strongest PNJ at the time the SSW was observed in ERA5). These composite maps of Z500 were computed for the period corresponding to the strongest burst of upward-propagating wave activity at 300 hPa in each case (3-13 Feb. 2018; 18-30 Dec. 2018; 1-10 Sep 2019). This approach allows us to assess whether the tropospheric state was different between the two groups of members and, consequently, whether it may have influenced the occurrence of SSWs. Thus, the plots for each model do not show the full Z500 pattern, as the ERA5 maps do, but rather the differences in tropospheric anomalies between the two groups of ensemble members. This explains, at least in part, why the amplitude of the model composite differences is weaker than that in the ERA5 plots and so, the colorbars should not be the same for ERA5 and the models.

Additionally, the comparison of the tropospheric circulation between the two groups of models helps identify the tropospheric structures that may modulate the likelihood of SSW occurrence in the models. However, in some cases, the stratospheric state between the two groups of ensemble members may not differ substantially as it happens, for instance, in GLOBO for SSW2018 or CNRM-CM6-1 for SSW2019 (Fig. 1e). As a result, even if the tropospheric pattern plays a key role in triggering the SSW, the corresponding anomaly maps may display lower magnitudes than those in ERA5.

In the revised version of the manuscript, the description of what Figures 5 to 7 show has been extended and clarified in L315-322 of the revised manuscript, explicitly highlighting the differences in the computation of ERA5 and model plots. The ensemble means of anomalous Z500 of the FREE simulation of each model have also been added to Fig. 5-7 in contours to help the interpretation of the mentioned difference maps. In those plots one can see that the magnitude of the anomalies is comparable to the ERA5 ones.

3) Figure 9: would it be helpful to add the multimodal mean?

The multimodel mean of HF at 100 and 300hPa for the three different experiments was already included in Figure 9 in gray. However, it was not clearly stated. In the revised version, we have included direct references to this multimodel mean in the figure caption and the associated description in the manuscript (L449-450).

4) Refractive index. The authors acknowledge around line 715 that the resonant growth is likely non-linear. It is my view that since the refractive index is derived from linear theory, it has serious limitations in how it can be applied and I would prefer if Fig 12 and 13 and associated discussion were omitted.

We thank the Reviewer for the suggestion. We have decided to omit the refractive index in Figures 12 and 13 and the associated description. Indeed, Section 5 has been simplified to bring the main points more clearly as also recommended by Reviewer 3.

5) The authors mention gravity wave drag as a source of uncertainty in models and as being important for triggering wave resonance. Have you looked at this in the SNAPSI models?

We agree with the Reviewer that examining the gravity wave drag would be highly valuable for the analysis. Unfortunately, only a limited number of the analyzed models provide gravity waves-related output variables. Specifically, three models include the eastward and northward acceleration by parameterized orographic gravity waves and only one model provides corresponding data for non-orographic gravity waves. Furthermore, two of these three models (GRIMs and CESM2-CAM6) are low-top models and are also those that present the largest challenges for representing wave propagation in the stratosphere during SSW2018. This is the reason why we propose this analysis for the future in the framework of a similar initiative as SNAPSI.

6) Line 745: I do not think interactive chemistry plays a role in the onset of SSWs but does later in the year. All the time periods analysed are in polar night.

We acknowledge that stratospheric ozone plays the most relevant role in shaping the polar stratospheric state after the polar night. However, several studies, including Haase and Matthes (2019) and Oehrlein et al. (2020), have shown that interactive chemistry also modulates the mean polar vortex state during the second half of winter (January-April), leading to a stronger vortex. Consistent with this strengthening, Haase and Matthes (2019) reported that interactive chemistry influences vortex variability, with a tendency toward fewer SSWs when chemistry-climate interactions are included. More importantly, both Haase and Matthes (2019) and Oehrlein et al. (2020) found that the time evolution of the polar temperature around the DJF SSW central dates differs depending on whether the interactive chemistry is considered or not. In particular, the inclusion of interactive chemistry tends to amplify the stratospheric warming associated with the onset of the SSW, likely due to increased dynamical heating linked to the greater wave activity necessary for an SSW to occur with a stronger mean vortex state. This is extremely important for the SSW forecast as it can determine whether a minor or a major SSW occurs.

These previous results suggest that interactive chemistry might play an active role in the onset of SSWs and therefore, represent an important aspect to investigate in future work.

Minor comments:

A general comment here is that the typesetting of the maths could be improved. For example, subscripts are used for both the z and \phi components of F and for partial derivatives.

The typesetting of the maths has been improved. In the new version the subscripts are restricted for the z and $\phi$ components of F.

Equation (1): Bold F for vector here and across manuscript. Added

Equations (1.1) and (1.2): \overline{\theta}_z rather than \overline{\theta_z} Modified

Line 190: Define z. Also de-italicize 'and \theta'. Done

Equation (2) actually comes from Kushner and Polvani (2004) Eq (7). The way they present it is much easier to read. Modified

Equation (4): Definition of q. Which PV? Quasi-geostrophic? Yes, we referred to the quasi-geostrophic vorticity. However, the equation has been removed in the new version of the manuscript following the recommendations of this Reviewer.

Table 3

   - First column. Consider using the event names introduced in Table 2. Modified

   - Second row: 2018 should be 2019.
   It referred to the first initialization of the FREE experiment for studying the SSW2019 that was on 2018-12-13. In any case, this has been replaced by the event names as suggested.

   - Third row. Maybe round 47.5% to 48% for consistency. Rounded

 Supplementary figures S2 to S4 are far too small.
Following Reviewer 2's suggestion we have included the ensemble means of anomalous Z500 of the FREE simulations in Figures 5-7 of each model. Thus, the new figures S2 to S4 only show the model means of this variable of the NUDGED and CONTROL simulations. Consequently, the figures look bigger than previously.

**References:**

Haase, S. and Matthes, K.: The importance of interactive chemistry for stratosphere–troposphere coupling, Atmos. Chem. Phys., 19, 3417–3432, https://doi.org/10.5194/acp-19-3417-2019, 2019.

Oehrlein, J., Chiodo, G., and Polvani, L. M.: The effect of interactive ozone chemistry on weak and strong stratospheric polar vortex events, Atmos. Chem. Phys., 20, 10531–10544, https://doi.org/10.5194/acp-20-10531-2020, 2020.

---

## Author Comment (AC2)

**REVIEWER 2**

General comments:

This study analyzed how the stratospheric condition influences the upward wave flux, and thus the occurrence of SSWs, based on a set of S2S model experiments in SNAPSI. The S2S forecasts are still struggling to predict SSW with long lead times. Thus, the current results provide helpful insights into understanding the role of stratosphere, which could potentially help improve the forecast skill in the future. The manuscript is very clean and well-written. However, since there are various diagnostic metrics and a huge amount of information, a major revision is needed to help with the presentation and delivery of the main conclusions.

Thanks a lot for your comments. In the revised version of the manuscript, we have worked to make the figures more informative and simplify, when possible, the discussion. In this sense, a large part of Section 5 has been rewritten and a better link has been established between model spread and model mean of the presented variables. Here is our reply to the major and minor comments in blue:

Major Comments:

1. Balance in presenting the model mean and model spread. Currently, most analyses are based on the model mean, while the model spread is not fully presented. For example, in Section 3, while Section 3.1 shows the mean skill of models in predicting zonal wind, Section 3.3 focuses on the difference in Z500 between 'weaker u' and 'strong u' groups. This 'inconsistency' poses a mismatch in the obtained information. It would be helpful to link the spread in Z500 with the spread in zonal wind. Thus, the model spread in predicting the zonal wind can be added in Fig. 1, and the model mean of Z500 should be included in Figs. 5-7. Although the fraction of members forecasting an SSW is shown in Table 3, this cannot accurately reflect the spread. For the latter, including the model mean will help in interpreting Figs. 5-7 as discussed in L360-365.

We thank the Reviewer for their valuable suggestion. In the revised version, we have included both types of information: the ensemble spread in zonal wind predictions and the ensemble mean of Z500 anomalies prior to SSWs.

Including the ensemble spread in u60_10 for each day and each model in Figure 1 is not straightforward as we have eight models plus two more ERA5 reference lines. Adding shadings to represent the ensemble spread would likely cause substantial overlap among them. However, we agree with Reviewers 1 and 2 on the necessity of illustrating the model spread for at least a relevant time period such as the dates around the occurrence of each SSW in ERA5 (12th-16th February 2018, 2nd-6th January 2019 and 18th-22nd September, same as those used to identify the "strongest u" and "weakest u" members). To do so, we have included three additional panels in Figure 1 (Fig.1d-f) showing the ensemble distribution of u60_10 for those dates. A brief discussion about the ensemble spread in each case has been added in the L251-260 of the revised manuscript that complements the analysis of the models forecast skill of the three SSWs.

As for the ensemble mean of the Z500 pattern, we have now included this mean for each model in contours in Fig. 5-7. This facilitates the interpretation of the spread in the results, as it allows direct comparison between the mean values and the differences between the two

groups of ensemble members ("weakest u" and "strongest u"). A corresponding discussion has been added accordingly.

Finally, we have connected the spread in Z500 with that in u60_10 by including, in each plot for Fig 5-7, the differences in the averaged u60_10 between "weakest u" and "strongest u" groups below each model name. This enables us to rapidly assess whether weak values in the Z500 difference patterns are due to low spread in both the stratosphere and troposphere, as it is the case for GLOBO during SSW2018, or whether the similarity is only restricted to the troposphere suggesting other processes may contribute to the deceleration of the vortex, as appears to be the case for most models during SSW2019 SH.

2. Discussion on the stratospheric state. While I understand that the main focus of this study is to provide a general understanding of the role of the stratospheric state, it would be worthwhile to add more discussion on what might constitute the stratospheric state. For instance, for the 2018SSW, the poleward shift of the PNJ is missing in the models, which is one possible candidate. In addition, the manuscript did not discuss about the QBO (although very briefly in L686), which is also included in the stratospheric condition and reflected by the experiment design. Although at such a short timescale the bias in the QBO may not be evident, the three SSWs occurred under different QBO states (e.g., Butler et al. 2020; Shen et al. 2020), and it would be worthwhile to reflect this and discuss the potential implications. In addition, as briefly discussed in Section 3.1, the intensity of the polar vortex relative to climatology also differs, which could also serve as preconditioning. The related discussion should be added to provide general implications.

Reference: Butler AH, Lawrence ZD, Lee SH, Lillo SP, Long CS. Differences between the 2018 and 2019 stratospheric polar vortex split events. Q J R Meteorol Soc. 2020; 146: 3503–3521. https://doi.org/10.1002/qj.3858

Shen, X., Wang, L., & Osprey, S. (2020). Tropospheric forcing of the 2019 Antarctic sudden stratospheric warming. Geophysical Research Letters, 47, e2020GL089343. https://doi.org/10.1029/2020GL089343

In the "Summary and discussion" section, we have emphasized what we define as stratospheric state, referring to both the polar vortex and the Quasi-Biennial Oscillation (QBO) and described it for the three analyzed SSWs in L690-712 of the revised manuscript.

Additionally, we have included brief references to the stratospheric state throughout the manuscript, particularly in the detailed analysis of SSW2018. In this case, as also suggested by the Reviewer, we have extended the figures showing the EP flux and wind distribution up to 10S (new Figure 12). This extension allows us not only to see more clearly the PNJ structure but also show the tropical winds, where an easterly phase of the QBO in the middle stratosphere is observed. A corresponding comment has been added in the figure description (L558-560 of the revised manuscript), emphasizing the role of equatorial easterly winds as a barrier to equatorward wave propagation and, consequently, contributing to the confinement of waves within the extratropics.

Nevertheless, we prefer to keep the discussion of the QBO influence on the predictability of SSW concise, as another SNAPSI working group is specifically dedicated to addressing this topic in depth.

3. Visualization of the plots. There are quite a few plots which are all informative. However, for some plots it is difficult to identify the regions/features being discussed in the main text. I suggest the authors adjust the plots to make the information more straightforward, which will help readers grasp the key information more quickly. Please see the detailed comments in the Specific Comments.

Following Reviewer's recommendation, most of the figures have been modified to make them more informative. These are the modifications:

- Figure 1: three new panels with the ensemble distributions of u60_10 averaged for the period 0/+4 days after the SSW in each model have been included.

- Figure 2-4: cyan solid lines representing NAVGEM results have been converted to dashed lines to easily distinguish them from those of CESM2-CAM6

- Figure 5-7: the ensemble mean of anomalous Z500 for the days preceding the SSW for each model has been added in contours. This helps in interpreting the difference in Z500 between the "weakest u" and the "strongest u" members. Finally, we have also added the multimodel mean of all these difference plots.

- Figure 8: the marker representing NAVGEM has been changed to distinguish it from CESM2-CAM6. Three more panels have been added to analyze the linkage between tropospheric circulation, upper tropospheric wave activity (HF300) and the lower-stratospheric wave activity (HF100). To do so, we show the pairwise correlations of the predominant WN HF100 and HF300, and predominant WN HF300 and the Z500 anomalies corresponding to the tropospheric precursors.

- Figure 11: dashed vertical lines of different colors have been added to delimit the periods of interest discussed in the text and a magenta solid vertical line has been included to indicate the central date of each SSW.

- Figure 12 and 13 have been merged to favor the comparison of zonal mean u and EP flux distribution in the two periods of interest ($3^{rd}$-$5^{th}$ Feb and $6^{th}$-$8^{th}$ Feb). The plots have been extended to 10ºS to see tropical winds configuration. Additionally, the refractive index has been removed from these figures to simplify the discussion.

- Figure 13 (old figure 14): a magenta solid vertical line has been added to indicate the central date of each SSW.

- Figure 14 (old figure 15) has been corrected as we found a small bug in WN2 Fz10 that also affected the net EP flux. The color of the median value line in each boxplot has also been changed to black to make it more visible. Further, the ERA5 reference line is now shown in magenta to distinguish it from the median value line and to ensure consistency with the boxplot formatting used in the other figures.

Specific Comments:

1. L210: Suggest explaining that the negative eddy heat flux indicates upward propagation in the SH. Despite the figure caption of Fig. 4, there is no explanation in the main text. Please add this.

We have added the following sentence in L212 of the revised manuscript: "HF is defined as positive for upward wave activity flux in both hemispheres"

2. L227: The classification of 'weakest u' and 'strongest u' is a bit counterintuitive, as the 'weaker' group corresponds to a better forecast, whereas the 'stronger' group corresponds to a worse forecast. Would it be better to define them as something like 'SSW-like' and 'no SSW', which is more straightforward?

We understand the Reviewer's concern. However, the suggested terminology ("SSW-like" and "no SSW") may be misleading, as in many cases, the "weakest u" members are still far from predicting an SSW, particularly, for the SSW2018. This can be better appreciated in the new panels d)-f) of Figure 1, which show the ensemble spread of u60_10 over the same periods used to identify the "weakest u" and "strongest u" ensemble members. For this reason, we prefer to retain the terms "weakest u" and "strongest u", but we have clarified their definition in the revised manuscript (L221-227).

3. L260, Figure 1: Suggest also showing the spread in predicted [u] as in my major comment 1.

Done. See the rest of our comments in the reply to the major comment 1.

4. L278: Suggest briefly stating that WN2 is mainly responsible for the 2018 SSW, otherwise it is a bit abrupt to focus on WN2 directly.

We added a clarification in L274-275 of the revised manuscript.

5. Figures 2-4: Since there are lots of lines, I suggest bolding the model being discussed in the main text for visualisation.

We agree with the Reviewer that many lines appear in the same figure. However, we believe that bolding the models discussed in the text could be misleading for the reader as bolding is typically associated with highlighting statistically significant results. Moreover, in many cases we refer to the general behavior across all models, so emphasizing only a subset of them may not be ideal and could hide the results of non-bolded models. To improve the interpretation, we have instead included the corresponding color line in the main text, when discussing results of specific models in L275-297 of the revised manuscript.

6. Figures 5-7:

• Suggest adding the variance of forecasted [u] among the members in the subtitles after the model's name. This could help provide a straightforward comparison and understanding of the linkage between the spread in tropospheric circulation and stratospheric response.

Since we have already shown the ensemble distribution of forecasted u60_10 in new panels d)-f) of Fig.1, we have added the difference in the mean of u60_10 between the "weakest u" and the "strongest u" members groups below the subtitles.

• Suggest also adding the box to indicate the region in ERA5, as the map is not very visual and thus takes time to identify the region of focus.

The boxes have been added.

• The climatological PWs are from ERA5. While I understand this is what one can do, it is also possible that the model bias in the climatological PWs can influence the interpretation of linear interference. Perhaps it would be better to include a brief discussion.

To avoid confusion, we have removed the climatological PWs from model plots. We have also added a warning when relating linear wave interference to the position of the Z500 anomalies in models with respect to the antinodes of climatological PWs in ERA5 in L368-369.

7. L348: Please clarify what 'this center of action' refers to.

Clarified

8. L380-410: Suggest checking the HF300 as well. As stratospheric wave forcing does not entirely originate from tropospheric forcing as discussed later on and shown in Yessimbet et al. (2022). It would be helpful to establish a linkage between the tropospheric circulation, the tropospheric wave forcing (HF300), and the stratospheric wave forcing (HF100).

Reference:
Yessimbet, K., Shepherd, T. G., Ossó, A. C., & Steiner, A. K. (2022). Pathways of influence between Northern Hemisphere blocking and stratospheric polar vortex variability. Geophysical Research Letters, 49, e2022GL100895. hRps://doi.org/10.1029/2022GL100895

We thank the Reviewer for this comment. We have examined the linkage between tropospheric circulation, the upper tropospheric wave forcing (predominant WN HF300) and the stratospheric wave forcing (predominant WN HF100). To do so, we compared the correlation of HF300-HF100 with the correlation of Z500-HF300 for each event and model. The values are now shown in the new panels d-f of Figure 8.

We indeed find interesting results that support our conclusions derived for the NH events. For instance, during SSW2019, there is a strong coupling between the tropospheric precursors and both HF300 and HF100, with very similar correlation values across models. This indicates that the troposphere played a dominant role in triggering this event. For SSW2018, although the correlations between Z500 and WN2 HF300 and between WN2 HF100 and WN2 HF300 are also high, they exhibit a wider spread than in SSW2019, particularly in the latter relationship. Together with the low correlation between WN2 HF100 and Z500, this suggests that the stratospheric wave activity during SSW2018 might be modulated by additional sources beyond the troposphere, at least in some of the models.

In the case of the SSW2019 SH, the results show that while the correlation between Z500 and HF100 is relatively high, the correlations involving intermediate variables, i.e. HF300-Z500 and HF300-HF100, are generally lower than in the NH events. A more detailed analysis would be required to fully understand this behavior, but it lies beyond the scope of the present study.

A detailed discussion of this result is included now in L406-430 of the revised manuscript.

9. Figures 8 and S1. I like Fig. S1, which is very informative. It not only confirms the linear relationships among the three variables but also indicates the spread among the model members. I suggest the authors move it to the main manuscript, perhaps merging it with Fig. 8 as they are related, and add more discussion on it. For instance, we can see that for SSW2018, the scatters are densely located in the upper level, indicating that the forecast [u] is overall quite strong and related to the weak HF (Fig. S1a). Moreover, for the HF and Z500, the scatters are located in the lower panel (Fig. S1d). Despite the weaker Z500-HF linkage, the Z500 is also quite diverse. For comparison, it would be better to use the same range for x and y axes across different cases, also add the corresponding ERA5 variables. This will also help to interpret the conclusion from Fig. 8.

We agree with the Reviewer that Fig. S1 is informative. However, following the revisions made in the manuscript, most of the information highlighted by the Reviewer, such as the ensemble spread in forecasted u60_10 and HF100 values for each model and SSW, is now presented in the boxplots of Figures 1 and 9, respectively, and discussed in the corresponding sections. Moreover, Figure 8 has been expanded in response to specific comment 8 of this Reviewer,

so adding these scatter plots to the main text would require an extra figure. Given that the number of figures in the manuscript is already high, we have decided to keep Fig. S1 in the Supplementary Material. We also prefer not to fix the scale of x and y axes to show the linear relationships among the three variables for all events. Otherwise, in some events, we could not see in detail the distribution of the values for the ensemble members. As suggested, the ERA5 variables have been added in the new version of the figure.

10. L432-437. Please briefly explain how the quantitative changes are computed.

We have just quantified the fraction (in %) of the multimodel mean of HF100 in the CONTROL run represented by the mean in the NUDGED experiment. A brief indication has been added, in L456-457 of the revised manuscript, the first time this is mentioned.

11. L440: It is interesting that for 2019SSW SH, although the models have relatively good performance in capturing the tropospheric wave forcing (i.e., for the multiEnsM, the observation is within the 1.5IQR), they fail to capture the observed stratospheric wave forcing. This seems to imply that the stratospheric wave forcing does not completely come from the tropospheric wave forcing. Whereas in the other two events, the skill for HF300 and HF100 is more similar.

Thank you for the comment. We agree that this is an interesting result. However, since this section focuses specifically on the comparison between NUDGED and CONTROL experiment results, we prefer not to include additional discussion here in order to maintain clarity and readability. We have noted this point, however, in the updated discussion of Figure 8, which now includes information on HF300 as well (L406-430 of the revised manuscript).

12. L477: Suggest adding 'for the majority of models' for clarity.

Added

13. L480: Suggest changing 'model simulates very similar anomalies to ERA5' to 'the ERA5 value lies within the IQR ..'

Changed

14. Figure 11: Suggest marking the period of interest.

We have added a solid vertical magenta line marking the central date of the SSW in ERA5, as well as four dashed lines (two yellow and two green) indicating the two periods of interest discussed in the text ($3^{rd}$-$5^{th}$ February and $6^{th}$-$8^{th}$ February).

15. L537-538: Suggest changing 'relatively weak wave activity that is simulated by reanalysis and models' to ' …wave activity seen in the reanalysis and simulated …'

The whole section has been modified, so this sentence has been removed.

16. Figures 12-13: Suggest adding the box to indicate the regions for EP flux budget and changing the y-axis labels to hPa for consistency with the main text. In addition, suggest extending the latitudes to 10S to show the QBO structure, related to my major comment 2.

We have modified these figures following the Reviewers' recommendations. In particular, the refractive index has been removed and Figures 12 and 13 have been merged and extended up to 10ºS. The corresponding description has been modified accordingly. The y-label of the

new Figure 12 has also been changed to hPa. The box has been added in Fig. S5 as it is the one showing the exact period used for the EP flux budget.

17. Figure 14: Please make sure the vertical magenta line mentioned in the figure caption is visible.

Thanks for the reminder. The vertical magenta line has been made visible in the figure.

18. Figure 15: Suggest changing the color of the median value line for visualization. In addition, while currently we can see the difference in Fz and Fy, we cannot find their relative contribution to the net EP flux convergence. According to the previous analysis (Figs. 12, 13), the difference in the horizontal wave forcing is evident, however, it does not appear to play an important role according to this plot. This poses a gap between these two sections. I suggest the authors add a scatter plot of Fz100 vs Fy for each member, similar to Fig. S1. This might help in understanding their relative roles.

We thank the Reviewer for this valuable suggestion. The change in the color of the median value line (also implemented) allowed us to identify a minor bug in the code affecting the WN2 Fz10 and consequently, the associated net EP flux. Although the main conclusions remain unchanged, the corrected figure provides an additional indication of wave resonance in the stratosphere. Specifically, we now see that both Fy and Fz10 increase in the NUDGED runs relative to CONTROL and even FREE, whereas the change in Fz100 across the three experiments is small in most models. This key point has been highlighted in the discussion of the figure.

The updated results are also consistent with the new Figures 12 and 13. The enhancement of Fy associated with the weaker vortex begins on 6$^{th}$ February in ERA5 and was already evident in new Figure 12f. The increase in Fz in the middle stratosphere in the NUDGED runs starts around 8$^{th}$-9$^{th}$ February, as shown in the new Figure 13. This cannot be detected in Figure 12f because it only covers 6$^{th}$-8$^{th}$ February. To provide a complete picture, we have added a new supplementary figure (Fig. S5) showing the EP flux and zonal wind distribution averaged over 6th-14th February, the same period used in new Figure 14 (previously Figure 15). The corresponding discussion has been updated to emphasize consistency with the earlier results.

To sum up, we conclude that the main contribution to the strong convergence of net EP flux before SSW2018 arises from the lower stratospheric upward-propagating wave activity and this term is not strongly affected by the prescribed stratospheric state. Nevertheless, we find indications that the zonally symmetric stratospheric state modulates the wave activity at higher levels, amplifying it in agreement with wave resonance phenomena, but only in high-top models.

Finally, following the Reviewer's suggestion, we include the scatter plots of Fz100 vs Fy for the three experiments (Fig. R2.1). While the CONTROL and particularly, the FREE, experiments display an approximately linear relationship, the NUDGED runs deviate from linearity and show a much larger spread. This behavior may reflect the presence of non-linear processes such as wave resonance growth, as discussed above. We have opted not to include this figure (Fig. R2.1) in the manuscript because the number of figures is already high, and the corrected Figure 14 has reduced the gap between the current section and the previous one.

[Figure]

**Figure R2.1**. *Scatter plot of the integrated $F_z$ at 100 hPa ($F_{100}$) vs $F_y$ at 55°N averaged during 6th-14th February 2018 for a) FREE, b) NUDGED and c) CONTROL experiments.*

---

## Author Comment (AC3)

**REVIEWER 3**

This study investigates the perfomance of 7 subseasonal-to-seaonal (S2S) forecast models in simulating 3 distint SSW events (2 in the NH and one in the SH). Moreover, it investigates the role of the stratosphere in triggering the Sudden Stratospheric Warmings (SSWs) in free-running simulations and two nudging setups where the stratosphere is either nudged to observations or climatology. The authors find that for some SSWs, the stratosphere plays a major role in modifying the stratospheric wave flux, but results seem to be very event-dependent. The paper is well written and results are presented clearly. The question posed is novel, since previous research has mainly focused on tropospheric drivers of SSWs. The paper is, however, very technical and I have some minor suggestions that migth improve readabilty. I recommend publication after the comments below have been addressed.

Thanks a lot for your comments. Here is our reply to the general and detailed comments in blue:

General comments:

1) Given that the main motiviation of the study is the improvement of predictability of SSWs, I think there is insufficient discussion on if and how the results presented help towards improving S2S forecasts, especially given that the mechanism/role of the stratopshere seems to be very event-dependent

We thank the Reviewer for this suggestion. We have expanded the discussion on the implications of our results for improving SSW predictability at the end of Section 6 (L740-759 of the revised manuscript), with particular emphasis on WN2 events, given the general low skill of models in forecasting this type of SSWs. In addition to the comments regarding WN2 events, we now also highlight that, although the mechanism and role of stratosphere are event—dependent, our results reveal that improved prediction of tropospheric circulation is key for increasing the forecast skill of all types of SSW. This is an important outcome of our analysis as it confirms the findings of Taguchi (2018). However, he could not isolate the potential influence of the stratospheric state on the upper troposphere and therefore, could not demonstrate that tropospheric precursors are even important for WN2 SSWs, where the perturbations are detected simultaneously throughout the whole atmospheric column (Esler and Matthewman, 2011). The SNAPSI experiments allow this isolation.

2) Especially section 5 is very technical and would benefit from a clear summary of the most important pints at the end of the section. It might even be shortened a bit to bring the main points across more clearly.

In the revised version, Section 5 has been rewritten and reduced. Following Reviewer's 1 recommendation, we have simplified the former Figures 12 and 13 (now combined into the new Figure 12) by removing the refractive index. Additionally, the description of this figure has been integrated with that of the previous Figure 11 to provide a more concise summary of the upward wave propagation in the FREE experiment.

Finally, when making some visual modifications in Figure 15 (now Figure 14), we have discovered a minor coding error that affected the Fz10 of WN2 component and consequently, the net EP flux associated with this wave component. Although the main conclusions remain unchanged, the corrected figure offers further evidence supporting the presence of wave

resonance in the stratosphere, thereby strengthening the connection between this subsection and the rest of the section.

Additionally, as suggested by this Reviewer, a very brief summary of the main points of this section has been added at the end of Section 5.

Detailed comments:

1) It is unclear to me how the boxes in Figs. 5-7 are derived. Please add a more detailed explanation of this. I think it would help to add the box also to the ERA5 panel.

The boxes indicate the tropospheric precursors of SSW or at least, regions associated with a weaker vortex state compared to the other states of the model ensemble. For each SSW, we define the areas characterized by substantial anomalies in the multi-ensemble mean of Z500 patterns (difference of Z500 between "weakest u" and "strongest u" ensemble members groups), as shown in Figs. 5-7 in shading. These areas are also located near the antinodes (ridge or trough) of the climatological WN1/WN2 waves in ERA5, suggesting that they could lead to constructive interference of waves if the locations of these antinodes in the models closely matched those in ERA5. Since this definition relies on the distribution of anomalies in the multi-model mean, we have added this multi-model mean to Figs. 5-7. The definition of the areas has been clarified in the text in L379-383 of the revised manuscript, and the corresponding boxes have also been added to the ERA5 panel.

2) Lines 375 ff.: I am not sure whether I understand how the Z500 anomalies are combined. In the text, it says "… *by computing the sum of averaged anomalies for centers with positive anomalies or positive-minus-negative…".* In the latter case, are you substracting the absolute mean value of the negative anomaly?

No. We are subtracting the negative anomaly, i.e. we are adding the absolute value of the negative anomaly, because we are computing the mean of the absolute values of the averaged anomalies for each center. We thank the Reviewer for this comment because we realized that the description of this calculation was incomplete in the original version of the manuscript and we have now clarified it in L381-383 of the revised manuscript.

3) Figure 8: Two models have almost same color (CESM2-CAM6 and NAVGEM). I suggest changing colors for better visibility.

We tested different colors but having so many models makes it difficult to select an additional one. Thus, for NAVGEM we have used a different marker in Figure 8 and dashed lines instead of solid lines in Figures 1-4.

4) Lines 446 ff.: In CNRM, not only an INCREASE in eddy heat flux at 100 hPa (Fig. 9a) is seen in NUDGED in the SSW 2018, but also a DECREASE in HF100 in this model in the SSW 2019 (Fig, 9b). Why? This should be discussed. The same tendencies can be seen in GLOBO and UKMO in the SSW 2019.

The Reviewer raised an important question. A short analysis has been performed to try to answer it. Based on linear wave theory, one might expect an increase in HF100 in NUDGED relative to the CONTROL run for the SSW 2019 as well. Figure 9 of the manuscript shows the total eddy heat flux at 100hPa (HF100) for all wavenumbers. However, when examining the heat flux for individual wavenumbers, it becomes clear that the decrease in HF100 in NUDGED

for CNRM-CM61, GLOBO, UKMO-GloSea6 and also CESM2-CAM6 during SSW2019 is mainly due to a reduction in WN1 HF100 (Figure R3.1a). Indeed, for WN1, the negative difference in HF100 between NUDGED and CONTROL runs in these models increases with respect to HF100 for all wavenumbers. Moreover, more models such as KMA-GloSea6, GRIMs, NAVGEM or IFS also show a negative difference in HF100 between NUDGED and CONTROL runs, when just isolating the WN1 wave activity. In contrast, if we look at the WN3 wave activity, we find opposite results (Figure R3.1b), with overall higher values of WN3 HF100 in NUDGED than in the CONTROL run. The latter result can be understood by the very weak PNJ in the few days preceding the wind reversal in ERA5 and thus in the NUDGED runs (see Figure 1b of the manuscript). These weak westerlies are weak enough to allow the propagation of WN3 wave activity. However, in the CONTROL runs the PNJ is stronger, making WN3 wave propagation difficult.

The differences in HF100 between NUDGED and CONTROL runs are only detected in the stratosphere, but not in the upper troposphere (HF300), where both experiments show very similar values of HF100 for all wavenumbers, WN1 and WN3.

Based on these results, we hypothesize that when the WN3 wave propagation in the stratosphere is favored by the stratospheric state, the propagation of WN1 wave activity is reduced. This is what we observe in the time evolution of WN1 HF100 and WN3 HF100 in ERA5 for this SSW too (Fig. 3 of the manuscript). There might then be a competition between WN1 and WN3 wave propagation in the stratosphere. In fact, Smith et al. (1984) also documented examples during the winter 1978/1979 when the amplitude of stratospheric WN1 wave displayed lower values, but WN2 and WN3 waves were enhanced. These authors linked it to the effects of wave-wave interactions. Similarly, Shi et al. (2017) also described interactions between WN3 and WN1 during the SSW 2005. We think that this is a really interesting topic that we should analyze in more detail as a follow-up analysis. Nevertheless, we have included a short comment about this in the L464-471 of the revised manuscript.

[Figure]

**Figure R3. 1** *Boxplots showing the ensemble distributions of (a) WN1 and (b) WN3 eddy heat flux (HF) at 100 and 300hPa and 45°-75° N for the period with the strongest value of HF preceding the SSW2019 for all models and multimodel mean (MultiEnsM), and the different experiments (FREE, NUDGED, and CONTROL runs). The interquartile range (IQR) is represented by the size of the box and the horizontal black line corresponds to the median value. Whiskers extend from the box to a distance of 1.5 times the IQR. Outliers (colored circles) are defined as points with values greater than 1.5 times the IQR from the ends of the box. ERA5 values are represented by horizontal magenta lines.*

5) Figure 10: Although the multi-model mean shows no difference between nudged and control, in many models there seem to be significant changes, but they do not agree on the sign (especially in the SSW2019 SH). I think it would be worth investigating/discussing this in more detail, as 5 out of 7 models show clear differences between nudged and control in Z500 impact for the SSW2019 SH.

We thank the Reviewer for pointing this out. In the original version of the manuscript, the differences between NUDGED and CONTROL runs in Z500 impact for the SSW2019 SH had not been highlighted because they are only significant in three out of the five models. Please note that we define significant differences when the interquartile range of the two distributions do not overlap.

However, although not always significant, all models except for NAVGEM tend to show a higher amplitude of the tropospheric precursors in NUDGED than in CONTROL runs. The main contribution to this difference comes from the Amundsen Sea Low (ASL) that tends to weaken more in the NUDGED experiment, as revealed by the composite maps of Z500 anomalies prior to the SSW (Fig. S4). Since a weak vortex projects on a weakening of the ASL (Turner et al. 2012) and the vortex is already weak well before the SSW in the NUDGED run (Fig.1c), we hypothesize that the troposphere is already affected by the weak polar vortex influence. This is also consistent with Jucker and Reichler (2023), who characterized the life cycle of SSWs in the SH. They found that the polar vortex starts to decelerate more than 50 days before the SSW date, and, consistently, a very strong weakening of the ASL and a generally negative phase of the Southern Annular Mode appears in the troposphere one month before the occurrence of the SSW.

Considering the agreement across models, we have briefly discussed these results in the revised version of the manuscript in L507-515 of the revised manuscript.

6) Lines 538 ff: I suggest changing the y-label in Fig. 12 to hPa instead of Pa to be consistent with the text. I also suggest marking the areas discussed in the text in Fig. 12 (e.g. 10-3 hPS). Otherwise the discussion surrounding this figure is hard to follow.

Figures 12 and 13 have been modified in the revised version of the manuscript. In particular, the refractive index has been removed and both figures have been merged and extended up to 10ºS. The corresponding description has been modified accordingly. The y-label of the new Figure 12 has been changed to hPa.

7) Lines 549 ff.: Again, it is difficult to follow the discussion here. Which "shift towards the pole"? Again, marking the corresponding areas in Fig. 12 would improve readability of this part.

As mentioned in the reply to the Reviewer's second major comment, Section 5 has been reduced and the description of the refractive index has been eliminated to improve the readability of this section.

8) Line 601: what is meant by "misrepresentation of the zonally symmetric stratospheric state in models"?

We refer to the differences in the representation of the zonally symmetric stratospheric state by the models with respect to ERA5 already shown in Figs. 11 and 12 and described in the corresponding text. We think that these differences may have contributed to the lack of a strong WN2 burst in the stratosphere since the $F_z$ shows larger values in the NUDGED runs, where these discrepancies have been eliminated, than in the FREE ones.

The sentences have been clarified in the revised version of the manuscript.

References:

Esler, J., & Matthewman, N. J.: Stratospheric sudden warmings as self-tuning Resonances. Part II: Vortex displacement events. J. Atmos. Sci., 68, 2505–2523, 2011.

Jucker, M. and Reichler, T.: Life cycle of major sudden stratospheric warmings in the Southern Hemisphere from a multimillennial GCM simulation. J. Clim., 36, 643-661, 2023.

Shi, C., Xu, T., Guo, D. and Pan, Z.: Modulating effects of planetary wave 3 on a stratospheric sudden warming event in 2005. J. Atmos. Sci., 74, 1549–1559, https://doi.org/10.1175/JAS-D-16-0065.1, 2017

Smith, A.K., Gille, J.H., and Lyjak, L.V.: Wave-wave interactions in the stratosphere: Observations during quiet and active wintertime periods. J. Atmos. Sci. 41, 363-373, 1984.

Taguchi, M: Comparison of subseasonal-to-seasonal model forecasts for major stratospheric sudden warmings, J. Geophys. Res.: Atmos., 123, 10231-20247, 2018

Turner, J., Phillips, T., Hosking, S., Marshall, G.J., and Orr, M.: The Amundsen Sea Low. Int. J. Clim., 33, 1818-1829, 2012.